# γ-Protocadherin structural diversity and functional implications

Kerry Marie Goodman[1,†], Rotem Rubinstein[1,2†], Chan Aye Thu[1], Seetha Mannepalli[1], Fabiana Bahna[1,3], Göran Ahlsén[2,3], Chelsea Rittenhouse[1], Tom Maniatis[1,4], Barry Honig[1,2,3,4,5*], Lawrence Shapiro[1,2,4*]

[1]Department of Biochemistry and Molecular Biophysics, Columbia University, New York, United States; [2]Department of Systems Biology, Columbia University, New York, United States; [3]Howard Hughes Medical Institute, Columbia University, New York, United States; [4]Zuckerman Mind Brain and Behavior Institute, Columbia University, New York, United States; [5]Department of Medicine, Columbia University, New York, United States

**Abstract** Stochastic cell-surface expression of α-, β-, and γ-clustered protocadherins (Pcdhs) provides vertebrate neurons with single-cell identities that underlie neuronal self-recognition. Here we report crystal structures of ectodomain fragments comprising cell-cell recognition regions of mouse γ-Pcdhs γA1, γA8, γB2, and γB7 revealing *trans*-homodimers, and of C-terminal ectodomain fragments from γ-Pcdhs γA4 and γB2, which depict *cis*-interacting regions in monomeric form. Together these structures span the entire γ-Pcdh ectodomain. The *trans*-dimer structures reveal determinants of γ-Pcdh isoform-specific homophilic recognition. We identified and structurally mapped *cis*-dimerization mutations to the C-terminal ectodomain structures. Biophysical studies showed that Pcdh ectodomains from γB-subfamily isoforms formed *cis* dimers, whereas γA isoforms did not, but both γA and γB isoforms could interact in *cis* with α-Pcdhs. Together, these data show how interaction specificity is distributed over all domains of the γ-Pcdh *trans* interface, and suggest that subfamily- or isoform-specific *cis*-interactions may play a role in the Pcdh-mediated neuronal self-recognition code.

*For correspondence: bh6@cumc.columbia.edu (BH); shapiro@convex.hhmi.columbia.edu (LS)

†These authors contributed equally to this work

**Competing interests:** The authors declare that no competing interests exist.

## Introduction

A characteristic of neural circuit assembly is that dendrites and axonal arbors of the same neuron do not stably contact one another, but are free to interact with the processes of other neurons (*Zipursky and Grueber, 2013*; *Zipursky and Sanes, 2010*). This fundamental property of neural circuit assembly is accomplished through a mechanism that mediates 'self-avoidance' between sister branches from individual neurons, while permitting interactions between non-self neurons. In both vertebrates and invertebrates, self-avoidance is thought to require the generation of unique single cell surface identities through mechanisms that involve the stochastic expression of unique combinations of cell surface protein isoforms (*Zipursky and Grueber, 2013*; *Chen and Maniatis, 2013*). In *Drosophila* and many other invertebrates individual-neuron identities are provided by the expression of single-cell-specific Dscam1-isoform subsets generated by stochastic alternative splicing (*Miura et al., 2013*; *Schmucker et al., 2000*; *Wojtowicz et al., 2004*; *Neves et al., 2004*; *Zhan et al., 2004*). By contrast, in vertebrates the clustered protocadherins (Pcdhs) provide analogous cell-surface diversity, but in this case generated through stochastic alternative promoter choice (*Tasic et al., 2002*; *Wang et al., 2002*; *Esumi et al., 2005*; *Hirano et al., 2012*; *Kaneko et al., 2006*).

Both the invertebrate Dscam1 proteins, and vertebrate Pcdhs are highly diverse families of cell-surface proteins that form isoform-specific *trans*-dimers between apposed neuronal cell surfaces (*Zipursky and Grueber, 2013*; *Zipursky and Sanes, 2010*; *Chen and Maniatis, 2013*; *Thu et al., 2014*; *Schreiner and Weiner, 2010*). Stochastic alternative splicing of the *Dscam1* gene in *D. melanogaster* produces up to 19,008 distinct protein isoforms, the majority of which engage in highly specific *trans* homodimerization (*Miura et al., 2013*; *Schmucker et al., 2000*; *Wojtowicz et al., 2004*, *2007*). In contrast, mice and humans express just 58 and 53 distinct Pcdh isoforms, respectively, each of which display isoform-specific homophilic binding in *trans* (*Schreiner and Weiner, 2010*; *Thu et al., 2014*). Biophysical measurements with domain-deleted proteins showed that Pcdhs also interact in *cis*, through a membrane-proximal dimer interface involving extracellular cadherin domain 6 (EC6) and potentially EC5 (*Thu et al., 2014*; *Rubinstein et al., 2015*). Pcdh *cis* dimers are thought to form promiscuously (*Schreiner and Weiner, 2010*; *Thu et al., 2014*), and thus provide a large repertoire of *cis* dimeric Pcdh recognition units (*Rubinstein et al., 2015*; *Thu et al., 2014*).

Vertebrate protocadherin genes have a unique organization in which the *Pcdhα*, *Pcdhβ*, and *Pcdhγ* gene clusters are arranged in tandem (*Wu and Maniatis, 1999*). Each of the Pcdh gene clusters contains multiple alternative variable exons (14 *Pcdhα*, 22 *Pcdhβ*, and 22 *Pcdhγ* in the mouse) which encode full Pcdh ectodomains, including six extracellular cadherin (EC) domains, a single transmembrane region and a short cytoplasmic extension. The *Pcdhα* and *Pcdhγ* gene clusters also contain three 'constant' exons that encode cluster-specific intracellular domains. The last two variable exons in the *Pcdhα* gene cluster and the last three variable exons of the *Pcdhγ* gene cluster are divergent from other Pcdh 'alternate' isoforms and are referred to as 'C-type' Pcdhs (*Wu and Maniatis, 1999*; *Wu et al., 2001*). The non-C-type *Pcdhγ* genes have been further divided into two subfamilies—*PcdhγA* and *PcdhγB*—based on sequence identity/phylogenetic analysis (*Wu and Maniatis, 1999*). Single-cell RT-PCR studies of the *Pcdhα* and *Pcdhγ* clusters in Purkinje neurons revealed that each neuron expresses all C-type Pcdhs biallelically, along with ~10 alternate isoforms (α, β, and γ) stochastically expressed from each gene cluster independently on allelic chromosomes (*Esumi et al., 2005*; *Kaneko et al., 2006*).

Each of the three Pcdh families may serve specialized functions. Knockouts of the *Pcdhα* gene cluster revealed neuronal wiring defects in olfactory and serotonergic neurons (*Hasegawa et al., 2008*, *2012*; *Katori et al., 2009*). By contrast, genetic ablation of the *Pcdhγ* gene cluster leads to lethality at P0 (*Lefebvre et al., 2008*; *Wang et al., 2002*), and revealed a cell-death phenotype for some neuron types (*Wang et al., 2002*; *Weiner et al., 2005*; *Lefebvre et al., 2008*; *Prasad and Weiner, 2011*; *Chen et al., 2012*). Conditional deletion of the *Pcdhγ* cluster which bypasses neonatal lethality, revealed defects in dendritic arborization of cortical neurons (*Garrett et al., 2012*). Similarly, *Pcdhγ* knockdown in hippocampal neurons in vitro resulted in dendritic arbors with lower complexity (*Suo et al., 2012*). Subsequent studies with transgenic and conditional knockout mice suggest that γ-Pcdhs act locally to regulate dendrite arborization, with the complexity of a neuron's dendritic arbor determined, at least in part, by Pcdh-dependent non-cell autonomous interaction of a neuron with surrounding neurons and glia (*Molumby et al., 2016*).

Clustered Pcdhs were first implicated in dendritic self-avoidance through studies of the *Pcdhγ* gene cluster. Deletion of all 22 genes in the *Pcdhγ* cluster in mice results in a loss of dendritic self-avoidance in retinal starburst amacrine cells and cerebellar Purkinje cells (*Lefebvre et al., 2012*), with formation of self-synapses (autapses) observed in starburst amacrine cells (*Kostadinov and Sanes, 2015*). However, most other neuron types appeared unaffected by the loss of the *Pcdhγ* gene cluster.

Cellular recognition specificities of Pcdhs appear to be diversified by co-expression of multiple Pcdh isoforms in the same cell (*Yagi, 2013*; *Schreiner and Weiner, 2010*; *Thu et al., 2014*). In general, recognition between cells expressing multiple Pcdhs is only observed when all expressed isoforms match. In early work, *Schreiner and Weiner (2010)* showed that expression of mismatched isoforms resulted in less binding between a cell population adhered to a surface and cells passed over them. We assessed the ability of cells co-transfected with up to five Pcdh isoforms to co-aggregate with cells containing various numbers of mismatches, and found that expression of even a single mismatch prevented co-aggregation in cell aggregation assays (*Thu et al., 2014*). Thus, even a single mismatched isoform is able to interfere with recognition. Importantly, this behavior—which we termed 'interference'—is not observed with classical cadherins (*Thu et al., 2014*). We therefore suggested that the interference phenomenon could arise from promiscuous *cis* dimerization

between co-expressed Pcdh isoforms to form single-cell repertoires of dimeric Pcdh recognition units (*Rubinstein et al., 2015*).

The specificity-determining cell-cell recognition interface of Pcdhs involves domains EC1–4, as shown experimentally through mutagenesis analysis (*Rubinstein et al., 2015*) and suggested by mutation correlation analysis (*Nicoludis et al., 2015*). Structures of the *trans* dimer formed through this interface have been reported for two α-Pcdhs and two β-Pcdhs which revealed overall-similar recognition-dimer structures, mediated by interfaces populated with diverse residue compositions that determine homophilic specificity (*Goodman et al., 2016*). Recently a *trans* dimer structure of a γB-Pcdh has also been reported (*Nicoludis et al., 2016*). Unfortunately, this structure contained a HEPES molecule in the EC2:EC3 interface, preventing formation of native specificity-determining intermolecular contacts in this region. Because a major portion of the binding interface adopts a non-native conformation, we have excluded this structure from subsequent analyses, unless specifically stated otherwise. Here we report structures of recognition dimers from four γ-Pcdhs, two from the γA subfamily, and two from the γB subfamily. The large collection of clustered Pcdh protein structures now available has allowed us to analyze the specificity determinants across the clustered Pcdh family.

In addition to new *trans*-dimeric structures, we also present the first Pcdh structures that include the promiscuous *cis*-dimerization region, although in monomeric form. Mutagenesis studies identify residues important for *cis* association and allow the visualization of these residues in the context of the structure. Finally, we show that Pcdh isoforms from the β, γA, and, γB subfamilies differ in their *cis* associations, and we report variability among homophilic *cis* associations of C-type Pcdhs. These differences, along with those previously characterized for α-Pcdhs (*Thu et al., 2014*), suggests that individual isoform- or subfamily-differences in *cis* interaction behavior may play an important in generating a Pcdh self-recognition code.

## Results

### *Trans* interactions and Pcdh specificity

Crystal structures of γ-Pcdh cell-cell recognition dimers

To characterize the cell-cell recognition (*trans*) interfaces of γ-Pcdhs we produced EC1–4 or EC1–5 fragments of mouse γA-, γB-, and γC-Pcdh isoforms using suspension HEK293 cells. These constructs encompassed the entire Pcdh EC1–4-mediated *trans* interface (*Rubinstein et al., 2015*; *Nicoludis et al., 2015*; *Goodman et al., 2016*), but lacked EC6, which mediates a distinct *cis* interface (*Thu et al., 2014*; *Rubinstein et al., 2015*). We used sedimentation equilibrium analytical ultracentrifugation (AUC) to characterize the homophilic binding properties of these proteins. The γA isoform constructs—γA1$_{EC1-4}$, γA4$_{EC1-4}$, γA8$_{EC1-4}$, γA9$_{EC1-5}$—displayed dimer dissociation constants (K$_D$s) of between 8.6 and 45.3 µM (*Table 1*). The γB isoforms (γB2$_{EC1-5}$, γB5$_{EC1-4}$, γB6$_{EC1-4}$, γB7$_{EC1-4}$) *trans* dimer affinities were more varied, with K$_D$s between 22 and 147 µM (*Table 1* and *Rubinstein et al. (2015)*). Finally, both γC isoform *trans*-interacting fragments tested—γC3$_{EC1-4}$ and γC5$_{EC1-5}$—formed relatively weak dimers, with K$_D$s of 115 and 100 µM respectively (*Table 1* and *Rubinstein et al. (2015)*).

Crystallization screening of these dimeric γ-Pcdh fragments yielded crystals of γA1$_{EC1-4}$, γA8$_{EC1-4}$, γA9$_{EC1-5}$, γB2$_{EC1-5}$, and γB7$_{EC1-4}$, and their structures were determined by molecular replacement (*Figure 1A* and *Figure 1—figure supplement 1A*). X-ray diffraction by the γA9$_{EC1-5}$ and γB7$_{EC1-4}$ crystal form 1 crystals was significantly anisotropic and therefore the data was truncated using ellipsoidal limits for structure determination and refinement (*Figure 1—source data 1* and *Figure 1—figure supplement 2*). The resolution of the final refined structures was 4.2 Å for γA1$_{EC1-4}$, 3.6 Å for γA8$_{EC1-4}$, 2.9/4.3/3.2 Å for γA9$_{EC1-5}$, 3.5 Å for γB2$_{EC1-5}$, 4.5/4.5/3.6 Å for γB7$_{EC1-4}$ crystal form 1, and 3.1 Å for γB7$_{EC1-4}$ crystal form 2. Data collection and refinement statistics are given in *Figure 1—source data 1*.

Each of the Pcdh crystal structures consists of seven-strand beta sandwich EC domains arranged end-to-end, as expected, with three calcium ions bound at each of the EC–EC junctions by canonical cadherin family calcium-binding motifs. The structures are decorated with both N-linked glycans and O-linked mannoses (*Figure 1A*), including two EC2 G-strand O-linked mannoses (residues 193, 194,

**Table 1.** EC1–4 is required for *trans* dimerization for all γ-Pcdh subfamilies. Oligomeric state and binding affinity of N-terminal Pcdh fragments in solution were determined by sedimentation equilibrium analytical ultracentrifugation. The ratio between the isodesmic constant ($K_I$) and dissociation constant ($K_D$) is given for cases where it is less than two, indicating possible non-specific binding.

| Pcdh fragment | Oligomeric state | Dissociation constant (μM) |
|---|---|---|
| γA1$_{EC1-3}$ | Monomer | N/A |
| γA1$_{EC1-4}$ | Dimer | 13.3 ± 0.93 |
| γA4$_{EC1-3}$ | Monomer | N/A |
| γA4$_{EC1-4}$ | Dimer | 45.3 ± 1.52 |
| γA8$_{EC1-4}$ | Dimer* | 30 ± 1.5* |
| γA9$_{EC1-5}$ | Dimer | 8.61 ± 0.35 |
| γB2$_{EC1-5}$ | Dimer | 21.8 ± 0.21 |
| γB5$_{EC1-4}$ | Dimer | 79.1 ± 4.3 |
| γB6$_{EC1-3}$ | Monomer | N/A |
| γB6$_{EC1-4}$ | Dimer* | 29 ± 4.9* |
| γB7$_{EC1-4}$ | Dimer | 146.7 ± 44.2 |
| γC3$_{EC1-4}$ | Dimer | 115 ± 1.49 ($K_I/K_D$ = 1.56) |
| γC5$_{EC1-3}$ | Monomer* | N/A |
| γC5$_{EC1-5}$ | Dimer* | 100 ± 4.33* |

*Data from **Rubinstein et al. (2015)**.

or 195 and 195, 196, or 197 in the various Pcdh structures), which appear to be conserved among clustered Pcdhs (*Rubinstein et al., 2015*; *Goodman et al., 2016*).

## Arrangement of γB-Pcdh *trans* dimers

The γB2$_{EC1-5}$ structure and both γB7$_{EC1-4}$ structures reported here each contain two molecules in the asymmetric unit, which are arranged as anti-parallel EC1–4 mediated dimers (*Figure 1A*), similar to those we previously observed for α- and β-Pcdhs (*Goodman et al., 2016*; *Figure 1—source data 3*). The two γB7$_{EC1-4}$ structures contain near identical *trans* dimers (root mean square deviation over aligned Cα atoms (RMSD) of 1.5 Å over 805 Cα's; *Figure 1—source data 3*), and the γB2$_{EC1-5}$ dimer is closely related to the γB7$_{EC1-4}$ dimers (RMSD ~3 Å; *Figure 1—source data 3*).

The γB2$_{EC1-5}$ and γB7$_{EC1-4}$ *trans* dimer structures show the same overall arrangement as the previously published γB3$_{EC1-4}$ *trans* dimer structure (*Nicoludis et al., 2016*). However, closer analysis revealed that the γB3$_{EC1-4}$ structure is an outlier among the γB structures, both in terms of its overall structure (*Figure 1—source data 3–7*), and in the interactions at the recognition interface (*Figure 1—figure supplement 3*). The γB3$_{EC1-4}$ structure contains a HEPES molecule in the EC2:EC3 interface that prevents proper engagement of EC2 and EC3 in the dimer (*Figure 1—figure supplement 3*). This results in far fewer residue contacts in the EC2:EC3 interface than have been observed for all other clustered Pcdh *trans* dimer structures (*Figure 1—figure supplement 3* and *Figure 1—source data 7*). This structure has therefore been excluded from our analysis of γB-Pcdh specificity, since many of the contacts in the recognition interface appear to be non-native.

## Flexibility in the arrangement of γA-Pcdh *trans* dimers

The γA structures showed an unanticipated variability in their molecular arrangement in the crystals. The γA1$_{EC1-4}$ crystal structure contained four molecules in the asymmetric unit: Two of which are arranged in an EC1–4-mediated antiparallel dimer, with all four EC domains involved in the dimer interaction (chains A and B); and two are arranged in an EC2–3-mediated antiparallel dimer, in which EC1 and EC4 are not involved in the dimer interaction (chains C and D) (*Figure 1A*). The EC2–3 portion of the dimer interaction is very similar between the two dimers in the structure (RMSD = 0.98 Å

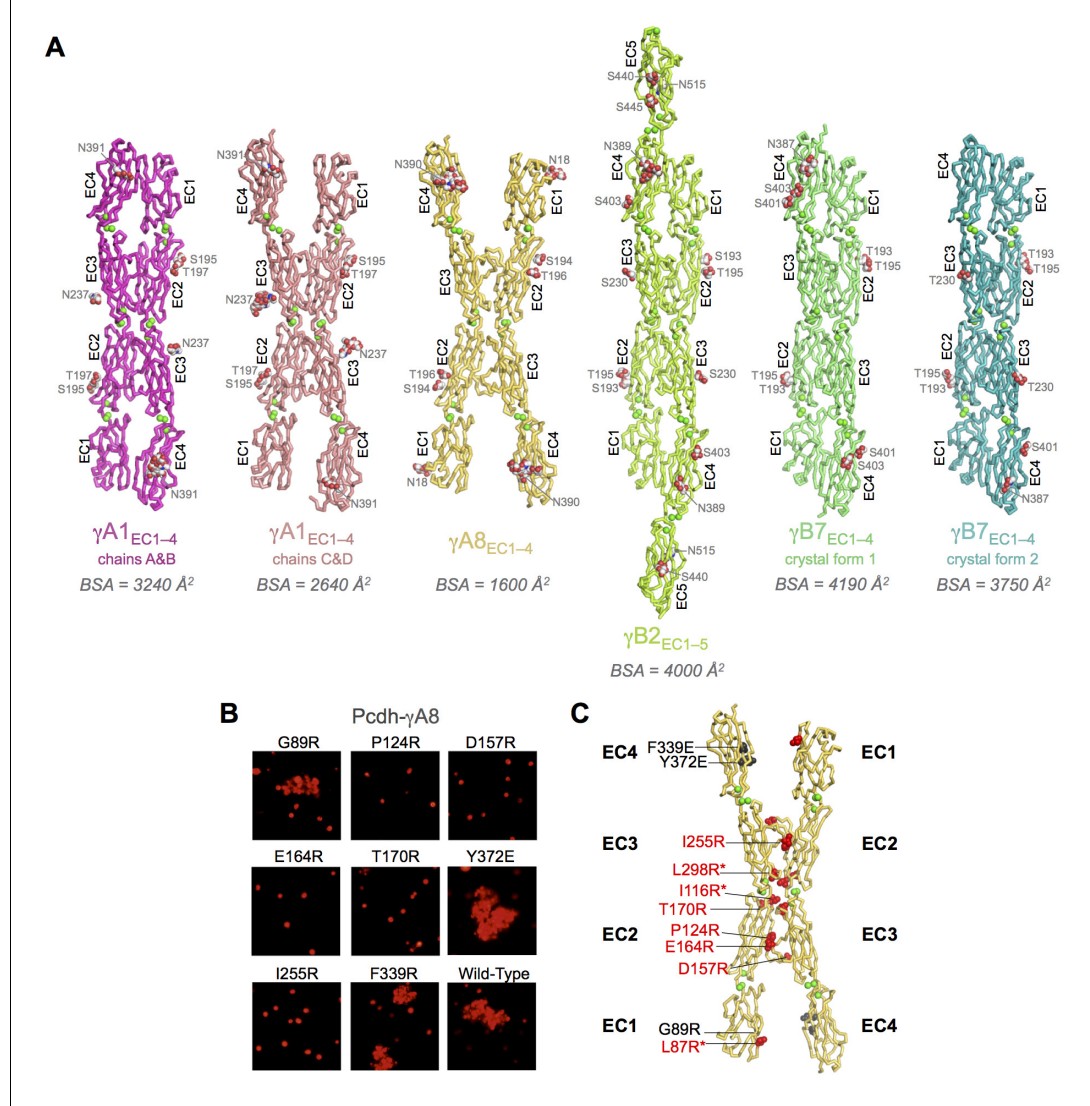

**Figure 1.** Crystal structures of γA- and γB-Pcdh cell-cell recognition dimers. **(A)** *Trans*-dimer structures of γA1$_{EC1–4}$, γA8$_{EC1–4}$, γB2$_{EC1–5}$, and γB7$_{EC1–4}$ fragments. The γA1$_{EC1–4}$ structure contained two distinct dimers in the asymmetric unit (chain A and B in magenta and chain C and D in salmon). The structures are shown in ribbon depiction with bound calcium ions shown as green spheres. Glycosylated residues are labeled, and glycans are shown as red, white and blue spheres. The buried surface area (BSA) in each dimer (see *Figure 1—source data 5*), given as the change in accessible surface area over both protomers, is shown beneath each structure. **(B)** K562 cell aggregation assays with γA8 mutants confirm the *trans*-dimer interface. **(C)** Mutations that prevent cell aggregation are shown on the γA8 dimer structure as red spheres and those which had no effect are shown as grey spheres. *Data from *Rubinstein et al. (2015)*.

The following source data and figure supplements are available for figure 1:

**Source data 1.** X-ray crystallography data collection and refinement statistics.
**Source data 2.** Overall structural similarity between EC1–4 regions of α-, β-, and γ-Pcdh structures.
**Source data 3.** Overall structural similarity between α-, β-, and γ-Pcdh EC1–4 *trans* dimer structures.
**Source data 4.** Overall structural similarity between α-, β-, and γ-Pcdh EC2:EC3 interface regions.
**Source data 5.** Overall structural similarity between α-, β-, and γ-Pcdh EC1:EC4 interface regions.
**Source data 6.** Pcdh protomer interdomain angles.

*Figure 1 continued on next page*

*Figure 1 continued*

**Source data 7.** *Trans*-dimer buried surface areas in all Pcdh EC1–4 containing crystal structures.

**Figure supplement 1.** γA9$_{EC1–5}$ monomer crystal structure and γA-Pcdh structural variability.

**Figure supplement 2.** X-ray diffraction anisotropy of the γA9$_{EC1–5}$ and γB7$_{EC1–4}$ crystals.

**Figure supplement 3.** Structural comparison of the EC2:EC3 interfaces observed in the γB3, γB2, and γB7 *trans* dimer structures.

**Figure supplement 4.** Mutagenesis experiments identifying the γA8 *trans* interface among the various crystal lattice contacts.

over 415 Cα's) and closely resembles the partial interaction observed in the previously published γA1$_{EC1–3}$ structure (*Nicoludis et al., 2015*; *Figure 1—figure supplement 1B*). The main difference between the two dimers in the γA1 crystal is therefore simply the presence or absence of the EC1: EC4 interaction. Since there are no protein domains filling the gap between EC1 and EC4 of chains C and D in the crystal, it is unclear why these domains do not interact. The fully engaged EC1–4-mediated dimer is similar to that of γB2$_{EC1–5}$ and γB7$_{EC1–4}$ and the published α- and β-Pcdh EC1–4-mediated dimers, involving the same interacting face of the molecule, however the RMSDs are quite large (4.3–5.0 Å; *Figure 1—source data 3*), highlighting the architectural differences between the γA1$_{EC1–4}$ dimer and those of other Pcdh subtypes (*Figure 1—source data 2–7*).

The γA8$_{EC1–4}$ crystal structure contained a single molecule in the asymmetric unit, which is engaged with a symmetry mate in an anti-parallel EC2–3-mediated interaction involving the same surface of the molecule as in the other clustered Pcdh *trans* dimer structures. This crystal also contained a distinct interaction between symmetry-related molecules, also mediated by an anti-parallel EC2–3 interface and with a similar buried surface area (*Figure 1—figure supplement 4*). In order to confirm which interface is the biological *trans* dimerization interface, we generated a number of γA8 arginine mutants separately targeting each of the observed interactions. Only those mutants that targeted the interaction surface in common with other Pcdhs resulted in loss of function in cell aggregation assays (*Figure 1B–C* and *Figure 1—figure supplement 4*). It is this γA8 dimer interaction that is shown in *Figure 1A*. Remarkably, like the γA1$_{EC1–4}$ dimer observed between chains C and D, the EC1:EC4 interaction is not formed. However, in the case of γA8 the interaction surfaces of EC1 and EC4 instead make contacts with the EC4 domain of another symmetry-related molecule in the crystal. These EC1:EC4 interactions are distinct from those observed in the fully engaged Pcdh *trans* dimers, although they utilize the same interaction surface of EC1. Since we have not observed molecular species larger than a dimer for Pcdh EC1–4 fragments we believe this alternative EC1:EC4 interaction, as well as the additional EC2:EC3 interaction, to be an artifact of crystallization.

Unexpectedly the γA9$_{EC1–5}$ crystal structure did not contain a *trans* dimer interaction in the crystal lattice (*Figure 1—figure supplement 1A*). Given that γA9$_{EC1–5}$ is a low micromolar dimer in solution (*Table 1*), the monomeric arrangement in the crystal is likely an artifact of crystallization, perhaps due to the low pH (6.5) of the crystallization condition.

Both the γA1$_{EC1–4}$ and γA8$_{EC1–4}$ crystal structures contain dimers mediated solely by the EC2–3 regions of the *trans* interface, suggesting that for γA-Pcdhs the EC2–3 interaction might be sufficient for dimerization and cell-cell recognition. To determine whether the EC2–3 regions are sufficient for dimerization of γ-Pcdhs we produced EC1–3 fragments of two γAs and a γB (γA1, γA4, and γB6). However, AUC of these fragments showed that all three were monomeric in solution (*Table 1*), like the EC1–3 fragments of β1 and γC5 (*Rubinstein et al., 2015*). Given these data we conclude that the EC1:EC4 interaction is required for dimerization. The absence of this interaction in some of the γA crystal structures is therefore likely an artifact of crystallization, perhaps due to crystallization condition, although it does imply that the EC1:EC4 interaction is not particularly stable.

## Inter-family specificity

To understand why members of the α-, β-, γA-, and γB-Pcdh subfamilies fail to form heterophilic cell-cell recognition complexes (*Thu et al., 2014*), we performed structural comparisons of the available homodimer structures. Excluding the structurally diverse γA-Pcdhs and the partially occluded γB3 dimer (*Nicoludis et al., 2016*), the EC1–4 dimers of isoforms from the same subfamily have similar overall structures (RMSDs ~1.5–3.4 Å; *Figure 1—source data 3*). This similarity is even more apparent when the two mutually exclusive interaction regions (EC1:EC4 and EC2–3:EC2–3) are compared separately revealing RMSDs of <2 Å (*Figure 1—source data 4–5*). In contrast, superpositions of dimers from different subfamilies in general revealed larger RMSDs due to distinct relative orientations of the individual protomers (>3.3 Å; *Figure 1—source data 3*; *Goodman et al., 2016*). These larger interfamily differences remain apparent when the interaction regions are compared separately, particularly for the EC2–3 dimer regions (*Figure 1—source data 4–5*). This, in itself, provides a simple explanation for the absence of α/β, α/γA, β/γA, and β/γB *trans* dimers. However, the five *trans* dimer structures from the α and γB subfamilies (excluding the structurally divergent γB3) exhibited intermediate structural similarity between the two subfamilies (RMSDs ~1.9–3.4 Å), which was even more apparent when the EC1:EC4 and EC2–3:EC2–3 interaction regions were compared separately (RMSDs <2.2 Å; *Figure 1—source data 4–5*). We therefore sought to identify other conserved elements that might distinguish these subfamilies, and distinguish γA- and γB-Pcdhs which are closely related in sequence.

The γB7 structure reveals a salt bridge in the EC1:EC4 interface between residues E41 in EC1 and K338 in EC4 (*Figure 2A*). Both E41 and K338 are conserved in all γB isoforms so that this salt bridge is likely present in all γB homodimers (*Figure 2C*). In addition, residue R340, which is also conserved in all γB isoforms, is positioned so that it could form an additional salt bridge with E41 (*Figure 2A*). In contrast, all γA isoforms have an arginine or lysine at position 41 (*Figure 2C*). Thus, a putative heterodimer formed between any γB isoform and any γA isoform would position a positively charged residue at position 41 in the EC1 domain of the γA isoform in close proximity to K338 and R340 in the γB isoform, which would significantly weaken binding (*Figure 2A–C*). Remarkably, α-Pcdhs also conserve a positive charge at position 40 (structurally equivalent to γB E41), which suggests that putative heterodimers between α-Pcdhs and γB-Pcdhs would also generate electrostatic clashes

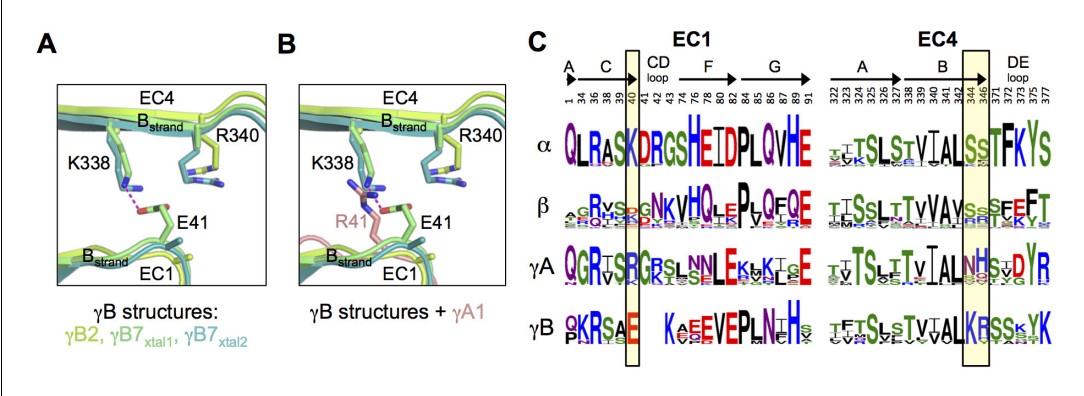

**Figure 2.** Pcdh interfamily specificity determinants in EC1 and EC4. (**A**) Close-up view of the interactions between E41, K338 and R340 (γB7 numbering) in the EC1:EC4 interface of the γB structures. (Side chains not present in the electron density maps of the respective structures were not built beyond the first carbon.) (**B**) Close-up view of a structural comparison between EC1 of γA1 (salmon) and γB structures. The interacting region in the γB EC4 domains are shown. While γB7 K338 forms a salt bridge with residue E41 in the γB7 homodimer, it would likely clash with γA1 R41 in a putative γA1:γB7 heterophilic complex. (**C**) Sequence logos of EC1:EC4 interfacial residues for each of the mouse Pcdh subfamilies excluding the divergent C-type isoforms (α, β, γA, and γB). The logos are generated from sequence alignments of all isoforms from each subfamily (see Materials and methods). Residue numbers correspond to Pcdhα7 numbering. Secondary structure elements are annotated above the logos. The black boxes highlight the sequence positions of residues participating in the γB7 EC1:EC4 salt-bridge interaction shown in A and B (E41, K338, and R340 in γB7 numbering).

The following figure supplement is available for figure 2:

**Figure supplement 1.** Sequence variability among Pcdh subfamilies in the interfacial regions of EC2 and EC3.

involving the same residues. Thus, the formation of heterodimers between γB-Pcdhs and both γA-Pcdhs and α-Pcdhs appears to be precluded by the conservation of key charged interface residues in EC1 and EC4. A similar mechanism has been shown to determine inter-family specificity in the desmosomal cadherins (*Harrison et al., 2016*) and intra-family specificity in the case of nectins (*Harrison et al., 2012*; *Samanta et al., 2012*).

## Intra-family γ-Pcdh *trans*-recognition specificity

We next considered *trans*-recognition specificity among γA isoforms and among γB isoforms. Our previous analysis of α- and β-Pcdhs showed that interfacial residues that vary between isoforms, yet are conserved in orthologs of a given isoform, function as specificity-determining residues (*Goodman et al., 2016*). Interactions between such residues were found to be favorable in homophilic complexes, but would typically generate steric or electrostatic clashes in potential heterophilic complexes. In order to identify specificity determining residues in γB and γA isoforms, we generated sequence logos derived from multiple sequence alignments of mammalian isoform-orthologs (*Figures 3* and *4*). The logo analysis revealed that the majority of isoform-specific *trans*-interface residues are highly conserved in the same isoform of other species.

To identify the likely roles of these residues in specificity we evaluated the relationship between residues that interact across the *trans* interface. For example, EC2:EC3 interacting residues 111 and 294 in the majority of γB2 orthologs (not including mouse) are lysine and aspartic acid, which are likely to form a salt bridge in the *trans* homodimer, whilst in γB3 and γB7 residue 111 is a conserved glutamic acid, which in the homodimeric complexes interacts favorably with H/R294 in γB3 or S294 in γB7. However hypothetical heterodimers between γB2 and γB3 or γB7 would juxtapose like charges E111 (γB3/7):D294 (γB2) and K111 (γB2):H/R294 (γB3) which would disfavor the heterophilic interactions (*Figure 3*, γB7 numbering). A similar example of electrostatic compatibility/incompatibility for homophilic/heterophilic pairing can be seen in the EC2:EC3 residues 128 and 257 in γA isoforms (*Figure 4*). We also found examples of small/large interacting residue pairs at the interface which showed correlated variations between isoforms such that heterophilic complexes would likely generate steric clashes: For example, EC1:EC4 residues 86 and 369 from γB isoforms, EC2:EC3 residues 125 and 253 of γB4 and γB5 (*Figure 3*, γB7 numbering), and EC1:EC4 residues 79 and 340 of γA8 and γA9 (*Figure 4*). Finally, we identified the self-interacting residue 206 of γA isoforms as a potential specificity-determining residue, providing hydrophobic contacts in some isoforms and polar contacts in others (*Figure 4*). In addition to such correlated variations in interacting residues, the logos also revealed conserved isoform-specific residues, which interact with residues that are conserved among γA or γB isoforms. Such residues may also contribute to specificity by favoring the homophilic interaction of one isoform over heterophilic interactions with other isoforms.

## *Cis* interactions

### EC6-dependent *cis* interactions of β- and γB-Pcdhs, but not γA-Pcdhs

We previously reported AUC data showing that γB6, αC2, and γC5 Pcdh EC1–6 fragments exist as dimers-of-dimers (tetramers) in solution, mediated by an EC1–4 interface and a distinct EC6-dependent interface (*Rubinstein et al., 2015*). Here, we have extended this analysis to determine the oligomeric states of multiple γ-Pcdh subfamily members and a representative of the β-Pcdh subfamily. All γA EC1–6 molecules we tested formed dimers rather than tetramers in solution (*Table 2*). γC3$_{EC1–6}$ was also a dimer in solution, although in this case the isodesmic constant was only 1.5 fold larger than the dimer dissociation constant, indicating non-specific binding. These EC1–6 dimers are mediated by an EC1–4 (*trans*) interaction, since all the γA and γC3 EC1–4 fragments we measured were also dimers in solution (*Table 1*) and the γA and γC3 EC2–6 or EC3–6 fragments were monomers or very weak non-specific dimers (*Table 2*). In contrast, γB2$_{EC1–6}$, γB4$_{EC1–6}$, γB5$_{EC1–6}$, γB6$_{EC1–6}$, γC5$_{EC1–6}$, and β5$_{EC1–6}$ were tetrameric in solution (*Table 2*). In addition, the γB2, γB5, and γB7 EC3–6 fragments were dimers in solution (*Table 2*), confirming the presence of the EC6-dependent *cis* interaction in solution for these γB- and β-Pcdhs, in contrast to the γA-Pcdhs. Since EC6 is highly conserved within non-C-type Pcdh subfamilies (*Table 3*), we assume these results will be general to all mouse β, γA, and γB isoforms.

We previously reported that γA8$_{EC2–6}$ was a dimer in solution (*Rubinstein et al., 2015*). However, it seems likely that this was due to the formation of an intermolecular disulphide bond

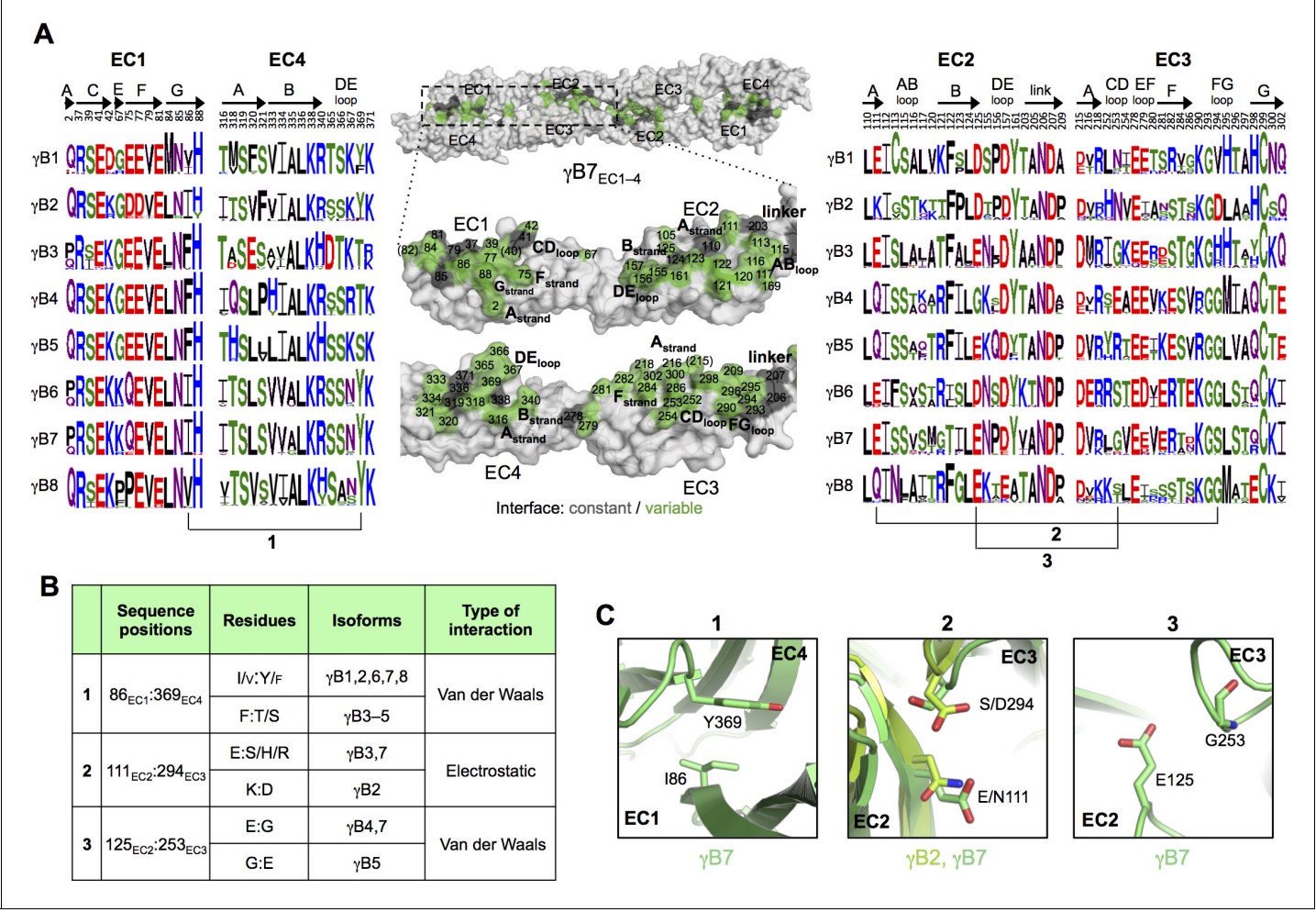

**Figure 3.** γB-Pcdh *trans*-binding specificity is encoded across the entire EC1–4 interface. (**A**) The central panel shows a surface view of the γB7$_{EC1-4}$ dimer, with half of the two-fold symmetric interface opened out to reveal the interacting faces. Interfacial residues are labeled and colored grey if they are constant among all γB isoforms or colored green if they vary among γB isoforms. The left and right hand panels show sequence logos for interfacial residues in EC1:EC4 (left) and EC2:EC3 (right) for each of the eight γB isoforms (NB γB3 is not present in mouse). The logos are generated from sequence alignments of multiple isoform-orthologs (see Materials and methods). γB7 residue numbering and secondary structure elements are annotated above the logos. The numbered connections between residue pairs correspond to the numbered rows in **B**. (**B**) Exemplar pairs of interacting residues that show conserved differences among a subset of γB isoforms and may therefore contribute to specificity. (**C**) Close-up views of the three interacting residue pairs highlighted in **B**. Residue pairs from the γB7$_{EC1-4}$ structure are shown in panels 1 and 3. Residue pairs from the γB2$_{EC1-5}$ and γB7$_{EC1-4}$ structure are shown in panel 2, since the identity of residues 111 and 294 varies between γB2 and γB7.

The following source data and figure supplements are available for figure 3:

**Source data 1.** List of species used in generating the sequence logos for γB-Pcdh isoforms.

**Figure supplement 1.** Sequence diversity of interfacial residues among γA and γB isoforms.

**Figure supplement 2.** γB7$_{EC1-4}$ dimer interface.

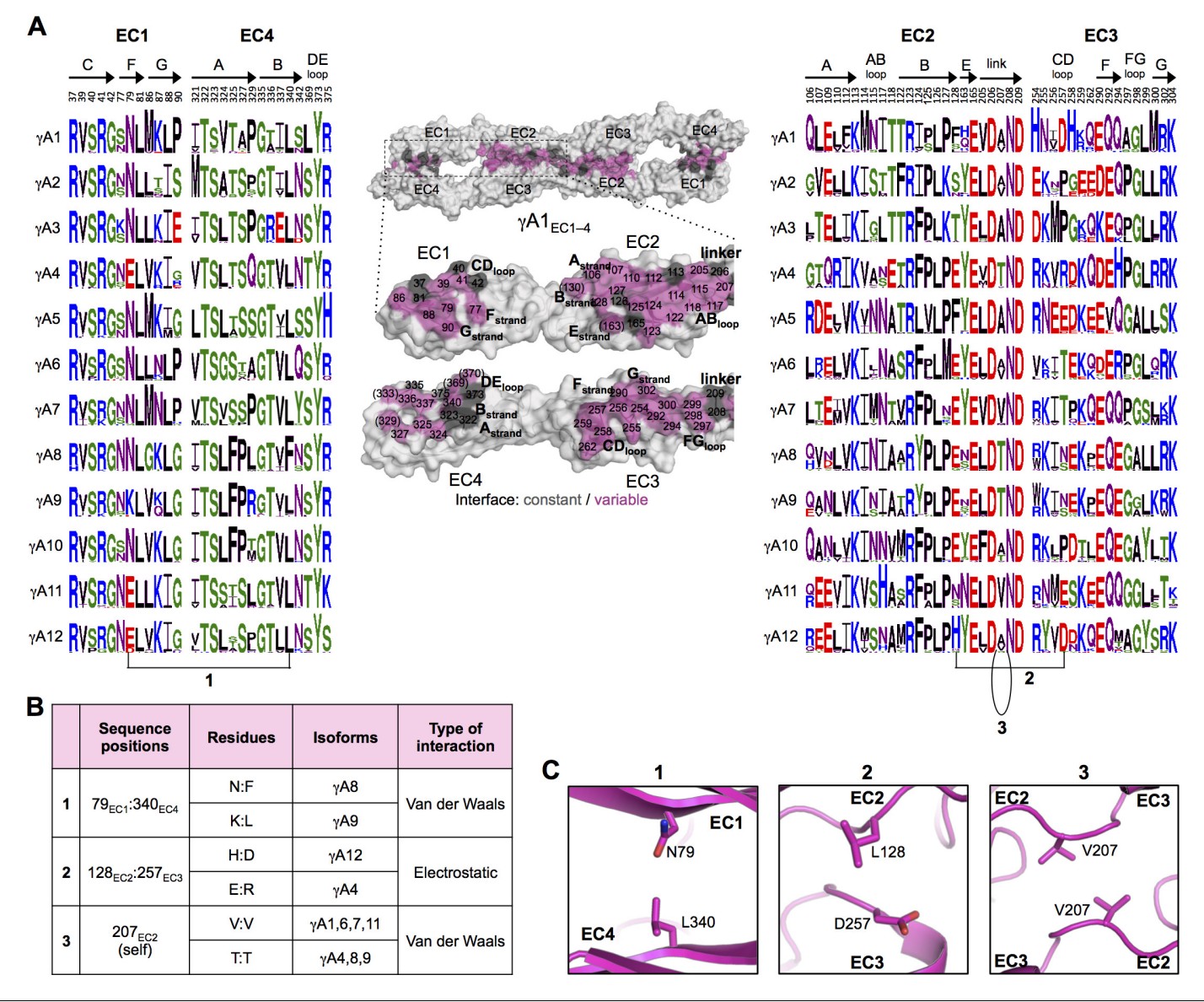

**Figure 4.** γA-Pcdh *trans*-binding specificity is encoded across the entire EC1–4 interface. (**A**) The central panel shows a surface view of the fully engaged EC1–4 γA1 dimer, with half of the two-fold symmetric interface opened out to reveal the interacting faces. Interfacial residues are labeled and colored grey if they are constant among all γA isoforms or colored magenta if they vary among γA isoforms. The left and right hand panels show sequence logos for interfacial residues in EC1:EC4 (left) and EC2:EC3 (right) for each of the 12 mouse γA isoforms. The logos are generated from sequence alignments of multiple isoform-orthologs (see Materials and methods). Secondary structure elements are annotated above the logos. The numbered connections between residue pairs correspond to the numbered rows in **B**. (**B**) Exemplar pairs of interacting residues that show conserved differences among a subset of γA isoforms and may therefore contribute to specificity. (**C**) Close-up views of the three interacting residue pairs highlighted in **B** are shown for the γA1EC1–4 structure.

The following source data and figure supplements are available for figure 4:

**Source data 1.** List of species used in generating the sequence logos for γA-Pcdh isoforms.

**Figure supplement 1.** γA1EC1–4 dimer interface.

**Figure supplement 2.** Experimental evidence for EC1:EC4 interactions contributing to Pcdh specificity.

**Table 2.** EC6-dependent homophilic *cis* interactions are observed for β- (orange rows), γB- (green), and some C-type Pcdhs (blue and purple) but not for γA-Pcdhs (yellow). Oligomeric state and binding affinity of Pcdh fragments in solution were determined by sedimentation equilibrium analytical ultracentrifugation. The ratio between the isodesmic constant ($K_I$) and dissociation constant ($K_D$) is given for cases where it is less than two, indicating possible non-specific binding.

| Pcdh fragment | Oligomeric state | Dissociation constant (μM) |
|---|---|---|
| **Entire ectodomains** | | |
| $β5_{EC1-6}$ | Tetramer | 3.9/3.2* |
| $γA1_{EC1-6}$ | Dimer | 1.18 ± 0.31 |
| $γA4_{EC1-6}$ | Dimer | 27.8 ± 0.73 |
| $γA9_{EC1-6}$ | Dimer | 7.81 ± 1.05 |
| $γB2_{EC1-6}$ | Tetramer | 2.8/8.9* |
| $γB4_{EC1-6}$ | Tetramer | 4.1/6.2* |
| $γB5_{EC1-6}$ | Tetramer | 3.4/1.3* |
| $γB6_{EC1-6}$ | Tetramer | 3.4/2.7* |
| $αC2_{EC1-6}$ | Tetramer[†] | 8.92/0.108*[†] |
| $γC3_{EC1-6}$ | Dimer | 61.6 ± 0.946 ($K_I/K_D$ = 1.51) |
| $γC5_{EC1-6}$ | Tetramer[†] | 18/7.64*[†] |
| $α7_{EC1-5}/γC3_{EC6}$ chimera | Tetramer | 3.0/3.9* |
| **Fragments containing the cis interaction region** | | |
| $γA1_{EC2-6}$ | Non-specific dimer | 403 ± 7.74 ($K_I/K_D$ = 1.15) |
| $γA4_{EC3-6}$ | Monomer | N/A |
| $γB2_{EC3-6}$ | Dimer | 80.1 ± 12.8 |
| $γB5_{EC3-6}$ | Dimer | 32.6 ± 4.6 |
| $γB7_{EC3-6}$ | Dimer | 59.0 ± 3.4 |
| $αC2_{EC2-6}$ | Dimer[†] | 8.92 ± 0.28[†] |
| $γC3_{EC3-6}$ | Monomer | N/A |
| $γC5_{EC2-6}$ | Dimer[†] | 18.4 ± 0.24[†] |

*$K_D$s of monomer-to-dimer / dimer-to-tetramer transitions from fitting the data to a tetramer model.
[†]Data from **Rubinstein et al. (2015)**.

mediated by an exposed cysteine residue, as was observed in the $γA8_{EC1-3}$ crystal structure (**Rubinstein et al., 2015**).

## γA-Pcdh carrier function suggests EC6-dependent heterophilic *cis* binding

We have previously shown that β17, γB6, αC2, and γC5 can interact with α-Pcdhs in an EC6-dependent manner (**Thu et al., 2014**). However, this has not been demonstrated for any γA isoform. Given

**Table 3.** Average pairwise amino acid sequence identities between EC6 domains of mouse Pcdh isoforms from each Pcdh subfamily.

| | Average pairwise sequence identity in EC6 |
|---|---|
| Alternate α-Pcdhs | 78% |
| Alternate β-Pcdhs | 90% |
| Alternate γA-Pcdhs | 90% |
| Alternate γB-Pcdhs | 96% |
| C-type Pcdhs | 45% |

the lack of a homophilic EC6-mediated homodimerization by γA isoforms in solution, we asked whether γA isoforms could interact heterophilically in *cis* with α-Pcdhs. To address this question, we performed the same assay as in *Thu et al. (2014)*, which depends on the observation that α-Pcdhs are not delivered to the cell surface when expressed alone in K562 cells, and are therefore not able to mediate cell adhesion. α-Pcdhs require co-expression of an EC5–6-containing fragment of a 'carrier' Pcdh from another subfamily to reach the cell surface and mediate cell adhesion. We therefore tested whether non-adhesive EC5–6 containing fragments of γA3 and γA9 were able to deliver Pcdhα4 to the cell surface to mediate cell adhesion. Co-expression of both these isoform fragments with Pcdhα4 resulted in cell aggregation (*Figure 5A*) indicating that, despite their apparent lack of homophilic *cis* dimerization, γA-Pcdhs can interact heterophilically with α-Pcdhs in *cis*.

## Crystal structures of γ-Pcdh EC3–6 fragments reveal the *cis*-interacting region

To further characterize γ-Pcdh *cis* interactions we sought to crystallize Pcdh fragments including both EC5, which may be involved in *cis* interactions, and the critical EC6 domain (*Thu et al., 2014*). From these experiments we obtained crystals of γA4$_{EC3-6}$ and γB2$_{EC3-6}$, which diffracted to sufficient resolution for crystal structure determination. X-ray diffraction by the γA4$_{EC3-6}$ crystals was significantly anisotropic (*Figure 5—figure supplement 1*), and therefore anisotropic resolution limits were applied. The resolution of the final refined structures was 3.0/4.3/2.85 Å for γA4$_{EC3-6}$ and 2.3 Å for γB2$_{EC3-6}$. Data collection and refinement statistics are presented in *Figure 5—source data 1*.

Both the γA4$_{EC3-6}$ and γB2$_{EC3-6}$ crystal structures consisted of four EC domains connected by linkers, each containing three bound calcium ions as expected (*Figure 5B*). The two structures are similar overall (RMSD of 3.02 Å over 405 Cα's), although γA4$_{EC3-6}$ shows a more pronounced EC4–EC5 bend angle (32.6° for γA4 vs. 18.6° for γB2). These are the first Pcdh structures containing EC6, which displays the classic beta sandwich fold, but with a large insertion between the A and A' strands (*Figure 5C*). This insertion is the one region of significant structural difference between the γA4 and γB2 EC6 domains, which otherwise have near identical structures (RMSD of 0.80 Å over 90 Cα's). Both structures are decorated with N- and O-linked sugar moieties throughout EC3–6, the majority of which are found at equivalent positions in both γA4 and γB2. Notably the G-strands of both EC6 domains are decorated with O-mannose groups on neighboring surface-facing residues, three for γB2 and four for γA4 (*Figure 5B*).

These EC3–6 structures, combined with the EC1–4/EC1–5 dimer structures, allowed us to model the EC1–6 *trans* dimer for γA- and γB-Pcdhs by structurally aligning the overlapping portions of the structures (*Figure 5D–E*). These models reveal an overall curved shape primarily defined by the EC4–5 bend angle, since both the EC1–4 dimer regions and the EC5–6 tails are relatively straight, and predict intermembrane spacings of ~360–375 Å.

The γA4$_{EC3-6}$ structure did not show any protein:protein interactions consistent with *cis* interactions in the crystal which, given that this γA family member is monomeric in solution (*Table 2*), was expected. However the γB2$_{EC3-6}$ crystal structure also did not reveal any interactions with clear biological relevance. Since γB2$_{EC3-6}$ forms a weak *cis* dimer in solution (80.1 µM; *Table 2*), this was unexpected. The monomeric arrangement in the crystal is likely an artifact of crystallization, perhaps due to the low pH of the crystallization condition (pH 6.5).

## Mutagenesis experiments and bioinformatics analysis reveal the Pcdh EC6 *cis*-interaction surface

In order to identify the clustered Pcdh *cis* interface we carried out mutagenesis experiments using γB6, which has been shown to interact both homophilically and heterophilically in *cis* and behaves robustly in cell aggregation assays and in biophysical assays (*Thu et al., 2014*; *Rubinstein et al., 2015*; *Table 2* and *Figure 5A*). We chose 11 EC6 surface residues, covering the entire surface of the domain, to mutate to aspartic acid. Wherever possible we chose residues that showed conserved differences between α-Pcdhs and other Pcdhs since it seemed likely that those residues are responsible for α-Pcdhs inability to form *cis* homodimers.

We first tested the ability of the mutants to deliver an α-Pcdh to the cell surface. To accomplish this, we produced all eleven mutants in a non-adhesive △EC1 γB6 context. We confirmed that these △EC1 mutants are non-adhesive in K562 cells when expressed alone, and then co-expressed each

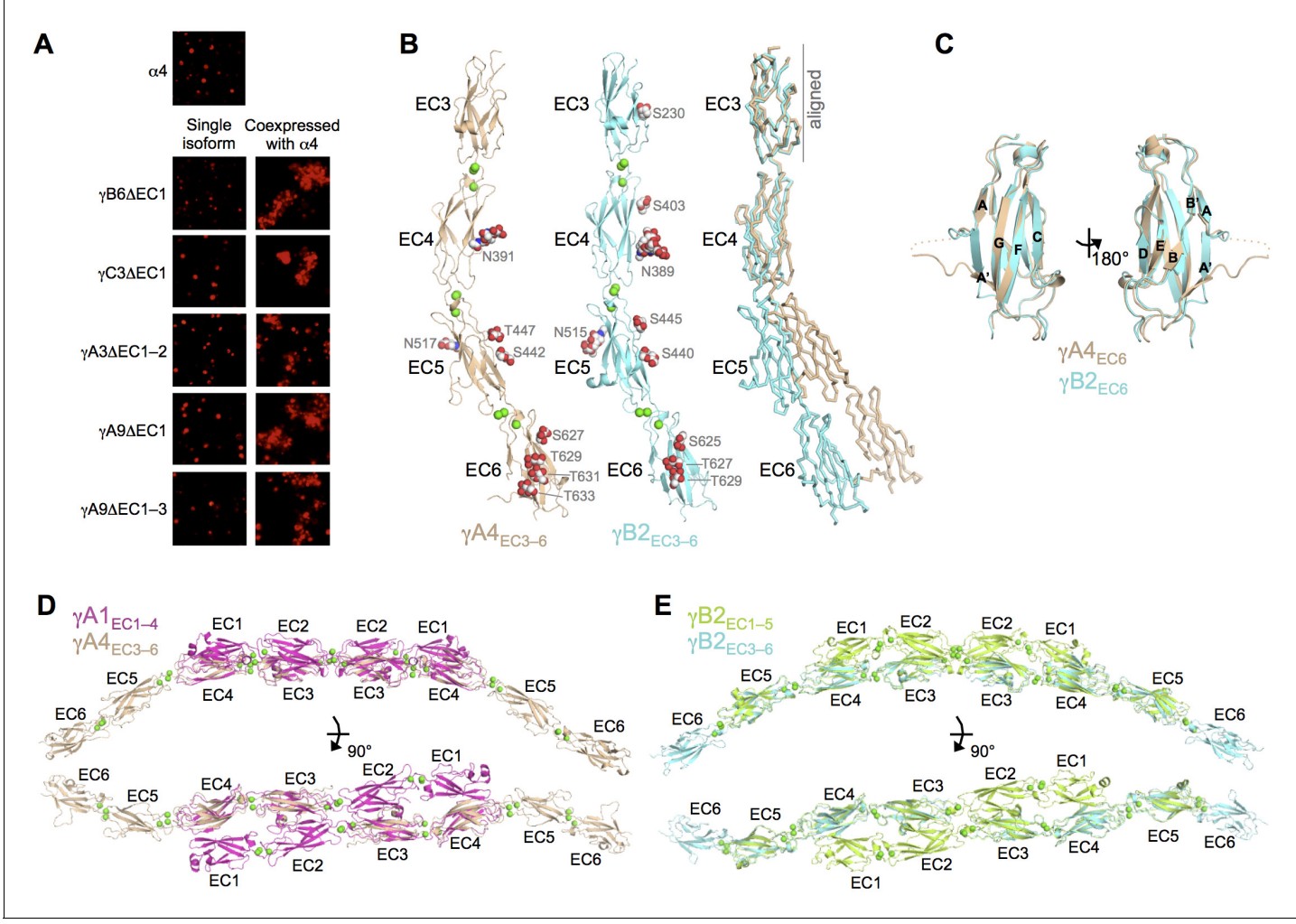

**Figure 5.** γA- and γB-Pcdhs can interact heterophilically in *cis* with α-Pcdhs and have similar *cis*-interaction region structures. (A) Cell aggregation assays with K562 cells. Pcdhα4 cannot mediate cell aggregation when expressed alone because it does not reach the cell surface (*Thu et al., 2014*; top panel). Additionally γ-Pcdhs lacking part of their EC1–4 *trans* interface also cannot mediate cell aggregation (left hand panels). However when these non-adhesive fragments of γA-, γB-, and γC-Pcdhs are co-expressed with full-length Pcdhα4, cell aggregation is observed (right hand panels). (B) Crystal structures of *cis* interaction region containing fragments of γA4 and γB2. Glycosylated residues are labeled and glycans are shown as red, white and blue spheres. Bound calcium ions are shown as green spheres. Structural alignment of the EC3 domains highlights the differences in curvature between the γA4 and γB2 EC3–6 fragments (right panel). (C) Structural alignment of the γA4 and γB2 EC6 domains reveals their near identical architecture. (D) Structural alignment of the overlapping EC3–4 regions of the γA1_{EC1–4} dimer with the γA4_{EC3–6} structure provides a model for the overall architecture of γA EC1–6 dimers. (E) Structural alignment of the overlapping EC3–5 regions of the γB2_{EC1–5} dimer with the γB2_{EC3–6} structure provides a model for the overall architecture of γB EC1–6 dimers. *Figure 5—figure supplement 1* and *Figure 5—source data 1*.

The following source data and figure supplement are available for figure 5:

**Source data 1.** X-ray crystallography data collection and refinement statistics for EC3–6 crystal structures.

**Figure supplement 1.** X-ray diffraction anisotropy of the γA4_{EC3–6} crystal.

mutant with an α-Pcdh to determine whether the α-Pcdh was successfully delivered to the cell surface, as indicated by whether the α-Pcdh could mediate cell adhesion. The majority of the γB6 mutants were able to deliver the α-Pcdh to the cell surface, but three mutants (L557D, V562D, and R597D, γB2 numbering) were not (*Figure 6A*). All three mutations mapped to the same surface of EC6, specifically to the B and E strands (*Figure 6B*).

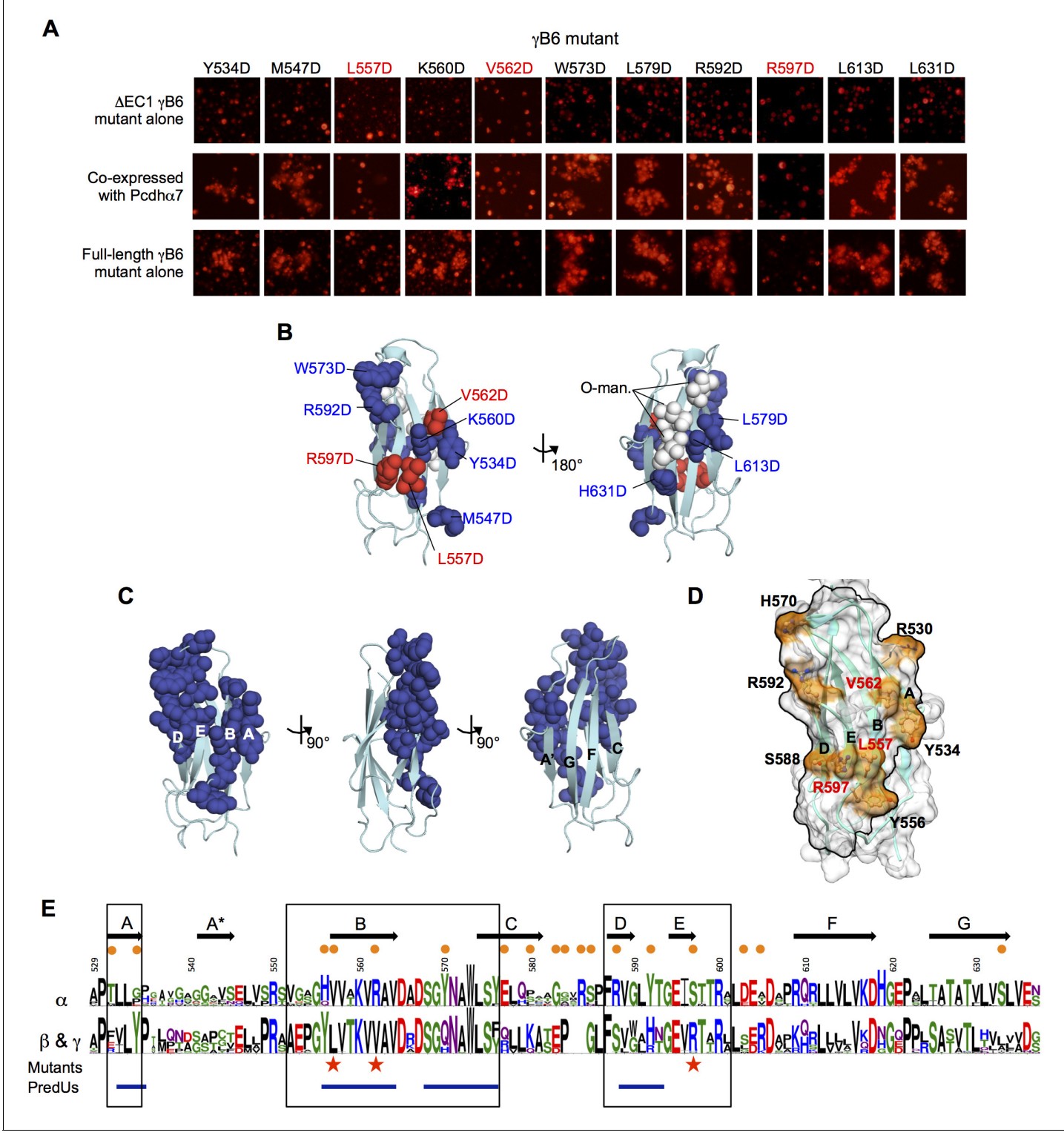

**Figure 6.** Identification and analysis of putative *cis* interface. (**A**) Probing the *cis* interaction interface by aspartic acid-scanning mutagenesis. Eleven EC6 point mutants of PcdhγB6 △EC1 (top panels) cannot mediate cell aggregation when expressed alone (top panel, γB2 numbering). When these 11 mutants are co-expressed with a full length Pcdhα7 cell aggregation is observed for eight of the mutants (middle panels), but not for the remaining three (highlighted in red). This is likely because of failure of these three mutant γB6s to carry Pcdhα4 to the cell surface. When assessed in the context of full-length PcdhγB6 expressed alone, the three mutants that were unable to deliver Pcdhα7 to cell surface did not aggregate cells, while the other eight mutants did mediate cell aggregation (bottom panels). (**B**) Residues mutated to aspartic acid are drawn in space filling representation on the γB2

*Figure 6 continued on next page*

*Figure 6 continued*

EC6 crystal structure. The three mutations that disrupted cell surface delivery of Pcdhα7 are colored red and the mutations that had no effect are colored blue. Glycans observed in the γB2 crystal structure are shown as white spheres and are found only on one side of the domain—the side opposite to the mutations disrupting cell delivery. (C) Residues predicted by PredUs2.0 (*Hwang et al., 2016*) to be interfacial are drawn in space filling representation on the γB2 EC6 crystal structure. Predicted interfacial residues occupy only one side of the molecule (composed of A, B, D, and E strands). This is the same side that was indicated by the mutagenesis approach to mediate *cis* interactions and opposite to the side that contains the glycans. (D) The predicted structure of γB6 EC6 (based on the γB2 structure) is shown in surface representation. Black lines frame the face of the molecule containing mutations that disrupt cell surface delivery (labeled in red) and the PredUs2.0 predicted interface residues. Nine surface-exposed residues that show different amino acid compositions between α-Pcdhs and the carrier β- and γ-Pcdhs are labeled and colored in orange. (E) Sequence logos of the EC6 domain for α-Pcdhs and all other alternate Pcdhs (β, γA, and γB). The logos are generated from sequence alignments of the mouse α1–12 isoforms and all mouse β, γA, and γB isoforms. Sequence positions that differ between α-Pcdhs and 'carrier' (β, γA, and γB) Pcdhs are highlighted by orange circles (above the logo); the three mutants that disrupt cell surface delivery are highlighted by red stars (below the logo); and regions predicted by PredUs to be in the interface are marked by blue lines (below the logo). Sequence positions within the face that is likely to contain the *cis* interface are boxed. Secondary structure elements are annotated above the logo.

The following figure supplements are available for figure 6:

**Figure supplement 1.** Analytical ultracentrifugation data of γB6$_{EC1-6}$ wild-type and V562D *cis* mutant.
**Figure supplement 2.** EC6 sequence analysis.

We also assessed the behavior of these mutants in the full-length γB6 context alone. While most were still able to mediate cell aggregation like wild type γB6 (*Thu et al., 2014*), the three mutants that were unable to deliver an α-Pcdh to the cell surface in the △EC1 context were also unable to the mediate cell aggregation in the full-length context (*Figure 6A*). Since all these mutations are in EC6 they should not affect the EC1–4-mediated *trans* interaction responsible for cell-cell adhesion in these assays. Thus, the fact that expression of these three mutants does not result in cell aggregation likely results from their failure to reach the cell surface.

To determine whether the L557D, V562D, and R597D γB6 mutants disrupt the *cis* interface, we attempted to express them in the EC1–6 context to assess their oligomeric state in solution by AUC. We were only able to produce one of the mutants, V562D. This EC1–6 mutant was a dimer in solution rather than a (*cis/trans*) tetramer like the wild type (*Table 4* and *Figure 6—figure supplement 1*), indicating the V562D mutation did indeed disrupt homophilic *cis* interactions. These results also suggest that, like the α-Pcdhs (*Thu et al., 2014*), cell surface delivery of γB isoforms requires EC6-mediated *cis* interactions.

Next, we used the PredUs2.0 program (*Hwang et al., 2016*), which combines structural homology with residue propensities, to predict EC6 surface residues likely to participate in *cis* interactions. Remarkably, all of the 23 residues predicted to be interfacial are located on one side of the molecule (*Figure 6C and E*)—the same side that was identified by mutagenesis. Furthermore, O-mannosylation is observed at EC6 G-strand residues 624, 626, and 628 (γB2 numbering) of both the γB2 and γA4 structures—on the opposite molecular face to the mutations that disrupt cell surface delivery (*Figure 6B*). These positions are usually conserved serines/threonines in α-, β-, and γ-Pcdhs (*Figure 6E*) suggesting that these O-glycans are likely present in all alternate Pcdhs.

**Table 4.** The γB6 mutant V562D disrupts the EC6-dependent *cis* interaction in solution. Sedimentation equilibrium analytical ultracentrifugation results for wild-type PcdhγB6$_{EC1-6}$ and the γB6$_{EC1-6}$ V562D (γB2 numbering) EC6 mutant.

| Pcdh fragment | Oligomeric state | Dissociation constant (µM) |
|---|---|---|
| γB6$_{EC1-6}$ | Tetramer | 3.4/2.7* |
| γB6$_{EC1-6}$ V562D | Dimer | 22.3 ± 0.793 |

*K$_{DS}$ of monomer-to-dimer / dimer-to-tetramer transitions from fitting the data to a tetramer model. Related to *Figure 6* and *Figure 6—figure supplement 1*

Together, these results allowed us to define a putative *cis* interaction region that encompasses the A, B, D, and E strands and the BC and DE loops of EC6 (*Figure 6D–E*). Sequence alignment of the EC6 domains for α, β, and γ isoforms shows that α-Pcdhs and the carrier β- and γ-Pcdhs differ in nine residues in this region (*Figure 6E*).

The structural basis for the differences in homophilic *cis* binding observed for γA and γB/β isoforms is not as clear. However, conserved sequence differences in the DE loop region between the γA, γB, and β subfamilies—γA = GLHT, γB = GLRT, and β = WAHN—as well as the top of the A strand (adjacent to B strand residue 562)—residues 531–532 are EI in γA, RV in γB, and FV in β isoforms—could contribute to the different subfamily *cis* interaction characteristics (*Figure 6—figure supplement 2*).

## Discussion

The structures of representative γA- and γB-Pcdh protein isoforms reported here complete a set of representative structures for *trans*-recognition interfaces from alternate clustered Pcdh isoforms, with structures now available for at least two Pcdhs from each of the α-, β- (*Goodman et al., 2016*), γB- (*Nicoludis et al., 2016*; this paper), and γA- (this paper) Pcdh subfamilies. Representative structures of engaged *trans* dimers of C-type Pcdhs have yet to be obtained. As discussed below, the collection of clustered protocadherin structures now available present a clear picture of how *trans*-homodimeric interaction specificity is coded for alternate Pcdh isoforms on the *trans* dimer interface comprising domains EC1–EC4. We also report monomeric structures of ectodomain regions containing the *cis*-interacting EC6 domain, and use them, together with mutagenesis experiments, to locate the *cis* interface in Pcdhs. In addition, our data indicate that γA and γB isoforms are distinct subfamilies with regard to their *cis* and *trans* protein interactions. With this information in hand, we discuss alternate mechanisms that have been proposed for the molecular basis of Pcdh-mediated neuronal self-recognition and non-self discrimination.

### Pcdh *trans* interaction specificity

The homophilic recognition properties of alternate (non C-type) clustered Pcdhs may be understood at the subfamily and isoform levels. Our data suggest that members of different subfamilies fail to bind to each other in *trans* primarily due to structural differences between the α, β, and γA subfamilies. That is, the putative dimers they would form would not exhibit shape compatibility. However, members of the γB subfamily are sufficiently similar in structure to members of the α subfamily that a specificity mechanism is unlikely to be based entirely on shape complementarity. Rather, the sequence and structural analyses presented above show that that EC1:EC4 interface in γB isoforms will contain salt bridges in the homodimers, whereas the comparable interaction in the inter-subfamily heterodimer would lead to incompatible electrostatic repulsion. In addition, electrostatic clashes involving the same residues appear to preclude formation of heterodimers between γB-Pcdhs and γA-Pcdhs. These then are cases where subfamily level specificity is encoded in the EC1:EC4 interface.

Sequence and structural analyses also identify the determinants of intra-subfamily specificity. In agreement with our previous analysis of the α- and β-Pcdhs (*Goodman et al., 2016*) we find that the electrostatic and steric compatibility apparent in homodimer structures would be replaced by incompatibility in putative heterodimers. As discussed above, some of the specificity-determining interactions are located in the EC1:EC4 interface and some in the EC2:EC3 interface. These findings, as well as those summarized in the previous paragraph contradict a primary conclusion reached by *Nicoludis et al. (2016)* that EC1:EC4 does not contribute to specificity. Based on their problematic structure of the EC1–4 *trans* dimer of PcdhγB3 (*Figure 1—figure supplement 3*) and on the four *trans*-dimeric α and β isoform structures we previously determined (*Goodman et al., 2016*), Nicoludis et al. used a bioinformatics analysis to infer that *trans* interaction specificity is mediated by the EC2:EC3 interaction, and that the EC1:EC4 interaction provides affinity, but not specificity. Our analysis, in contrast, reveals numerous specificity elements in EC1:EC4 interactions.

The importance of the EC1:EC4 interaction to *trans*-binding specificity is also demonstrated by our previously published experimental results with Pcdh mutants (*Figure 4—figure supplement 2*). Cell aggregation experiments with domain-shuffled mutants have clearly demonstrated that specificity is dependent on the identity of EC1 and EC4 (*Figure 4—figure supplement 2*; Figure S3 in

*Rubinstein et al. (2015)*). In the case of α-Pcdhs, it is true that the EC1 and EC4 interface residues are mainly conserved between isoforms, as we previously reported (*Goodman et al., 2016*). However, some isoforms show conserved differences that determine specificity: α7 shows isoform-specific conservation of *trans*-interface EC1 residues 36 and 38 and EC4 residues 322 and 324. Most importantly, swapping these residues between α7 and α8 swaps their binding preferences (*Figure 4—figure supplement 2*; Figure 5 in *Rubinstein et al. (2015)*). For β-, γA-, and γB-Pcdhs, isoform-specific conservation of EC1:EC4 *trans*-interface residues is observed in almost all isoforms, as can be seen in sequence logo analysis (*Figure 3*; *Figure 4*; and Figure 2 in *Goodman et al. (2016)*). Contrary to the conclusion reached by *Nicoludis et al. (2016)*, these observations, in addition to results from functional mutagenesis experiments showing that changes in specificity result when such residues are mutated (*Figure 4—figure supplement 2*; Figure 5 in *Goodman et al. (2016)*), clearly demonstrate that both the EC2:EC3 and EC1:EC4 interfaces play important roles in determining binding specificity.

## EC6 domain structure and *cis* interactions

Pcdh *cis* multimers have been suggested to form promiscuously between isoforms, and to thereby diversify the functional Pcdh repertoire (*Schreiner and Weiner, 2010*; *Yagi, 2013*; *Thu et al., 2014*; *Rubinstein et al., 2015*). We previously used domain-deletion studies of numerous Pcdh isoforms to localize the *cis* interaction region to the EC6 domain, with possible contributions from EC5, and showed that the *cis* complexes formed are dimeric (*Rubinstein et al., 2015*). Here we report structures containing monomeric EC6 domains, and locate their dimeric recognition regions by identifying mutations that interfere with the formation of *cis* dimers for both α- and γB-Pcdhs (*Figure 6*).

Sequence comparisons of Pcdh EC6 domains (*Figure 6* and *Figure 6—figure supplement 1*) revealed conserved differences between the Pcdh subfamilies, which are likely to relate to their *cis*-interaction specificities (*Thu et al., 2014*, *Rubinstein et al., 2015*). We previously showed subfamily specific diversity in *cis* interactions in that α-Pcdhs and PcdhγC4 are not transported alone to the cell surface, but only when engaged in *cis*-dimeric complexes with 'carrier' Pcdhs corresponding to other isoforms, including alternate β and γ, and some C-type Pcdhs (*Thu et al., 2014*). Our results suggest additional diversity in Pcdh *cis* interactions: we found through biophysical measurements that five alternate γB-Pcdhs interacted homophilically in cis in solution (*Table 2*), but three alternate γA-Pcdhs did not. In light of the high level of sequence conservation of the EC6 domains within the γA and within the γB-Pcdh subfamilies (*Table 3* and *Figure 6—figure supplement 2*), it is likely that, in general, γA-Pcdhs fail to dimerize or form only weak *cis* dimers (enabled in part by the constrained 2D environment of the membrane surface; *Wu et al., 2011*), while alternate γB/γB *cis*-dimers are expected to have significant affinity. Despite the difference in homodimerization affinities, both the γA- and γB-Pcdhs functioned as carriers for α-Pcdhs (*Figure 5A*), consistent with the participation of both γA and γB isoforms in Pcdh *cis* heterodimers. Overall, these observations clearly show an unanticipated specificity in *cis*-dimer formation.

## Implications for neuronal recognition

Subfamily specific differences in *cis*-dimerization specificity are expected to impact the diversity and composition of the functional Pcdh repertoire of *cis*-dimeric recognition units. It has previously been assumed that the *cis*-associated recognition units are composed of random isoform combinations (*Yagi, 2013*; *Thu et al., 2014*; *Rubinstein et al., 2015*). However our data suggest that the *cis*-dimer repertoire will not be random. For example, no recognition units consisting of two alternative α-Pcdh isoforms are expected to form, and γA/γA recognition units would be absent or less frequent than γB/γB recognition units. Since the composition of the *cis*-dimer repertoire is therefore limited compared to all random combinations, the recognition-unit diversity encoded by stochastic expression of Pcdh isoforms is likely to be less than previously thought.

We have previously described two alternative molecular mechanisms for neuronal self-recognition through *trans* interactions of Pcdh *cis*-dimeric recognition units (*Thu et al., 2014*; *Rubinstein et al., 2015*). Both of these mechanisms depend on diverse repertoires of dimeric recognition units. In the first case (*Figure 7B*), *trans* binding is envisioned to occur only between recognition units with precisely matched isoform composition, and results in the formation of a dimer-of-dimers containing maximally two Pcdh isoforms. As we described previously (*Thu et al., 2014*), this model leads to a

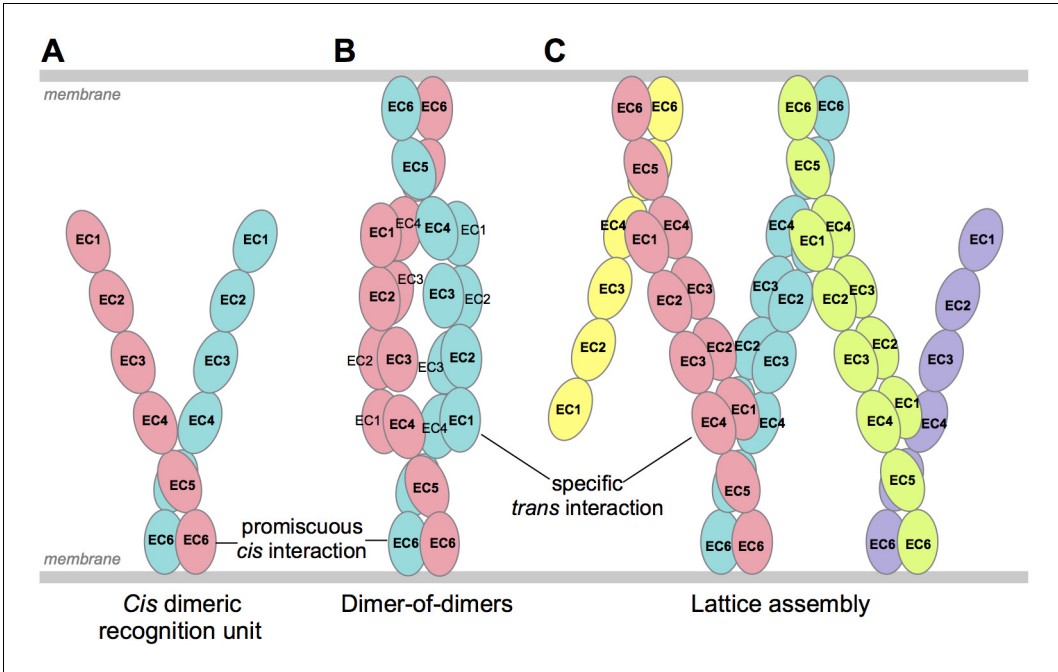

**Figure 7.** Schematic of possible modes of Pcdh-mediated recognition. (A) Pcdhs form homophilic and/or heterophilic *cis*-dimers, which are thought to be the recognition unit. (B) One possible Pcdh recognition complex is a dimer-of-dimers, which has been observed in solution for homophilic complexes of β, γB, and some C-type Pcdhs (*Table 2*). In this model the specificity of the *trans*-interaction would require both arms of the *cis*-dimer to match for recognition (*Rubinstein et al., 2015*). (C) An alternative recognition complex that has been proposed is a linear zipper (*Rubinstein et al., 2015*). In this model only one arm of opposing *cis*-dimers needs to match to join the assembly, but incorporation of a dimer containing an isoform that is not expressed by the opposing cell would terminate growth of the intercellular Pcdh zipper.

limited number of possible distinct cell surface identities and even fewer if the population of *cis* dimers is not random. In the second case (*Figure 7C*), *trans* binding is suggested to occur between recognition units with a single matched isoform, resulting in the formation of a zipper or chain of Pcdh dimers arrayed between membrane surfaces (*Thu et al., 2014*); the chain of dimeric recognition units is proposed to be terminated by the presence of a single mismatched isoform (*Figure 7C*). This chain termination model leads to the ability to encode a far larger set of distinct cell surface identities (*Rubinstein et al., 2015*). However, to date there has been no direct observation of oligomeric Pcdhs on cell surfaces. Since it remains unclear what proportion of neurons utilize Pcdhs for self-avoidance, and thus the Pcdh diversity required to avoid inappropriate self-recognition of interacting neurons remains unclear, we cannot currently distinguish between these models.

## Materials and methods

### Protein production

cDNAs for Pcdh ectodomain fragments, excluding the predicted signal sequences, were cloned into a pαSHP-H mammalian expression vector (a kind gift from Daniel J. Leahy, John Hopkins University) modified with the BiP signal sequence and a C-terminal octahistidine tag (*Rubinstein et al., 2015*). The signal sequences were predicted using the SignalP 4.0 server (*Petersen et al., 2011*).

Suspension-adapted HEK293 Freestyle cells (Invitrogen) in serum free media (Invitrogen) were used for protein expression. The plasmid constructs were transfected into cells using polyethyleneimine (Polysciences Inc.). Media was harvested ~6 days after transfection and the secreted proteins were purified by nickel affinity chromatography followed by size exclusion chromatography in 10 mM Tris pH 8.0, 150 mM sodium chloride, 3 mM calcium chloride, and 200–250 mM imidazole pH

8.0. Purified proteins were concentrated to >2 mg/mL and used for analytical ultracentrifugation or crystallization experiments.

## Sedimentation equilibrium analytical ultracentrifugation (AUC)

Experiments were performed in a Beckman XL-A/I analytical ultracentrifuge (Beckman-Coulter, Palo Alto, CA), utilizing six-cell centerpieces with straight walls, 12 mm path length and sapphire windows. Samples were dialyzed overnight and then diluted in 10 mM Tris pH 8.0, 150 mM NaCl, 3 mM $CaCl_2$ with varying concentration of imidazole pH 8.0, as follows: 200 mM ($\beta5_{EC1-6}$, $\gamma A1_{EC3-6}$, $\gamma A4_{EC1-4}$, $\gamma B2_{EC1-5}$, $\gamma B2_{EC1-6}$, $\gamma B2_{EC3-6}$, $\gamma B4_{EC1-6}$, $\gamma B5_{EC1-4}$, $\gamma B5_{EC1-6}$, $\gamma B5_{EC3-6}$, $\gamma B7_{EC1-4}$, $\gamma B7_{EC3-6}$, $\gamma C3_{EC1-4}$) or 250 mM ($\gamma A1_{EC1-4}$, $\gamma A1_{EC2-6}$, $\gamma A1_{EC1-6}$, $\gamma A4_{EC1-3}$, $\gamma A4_{EC3-6}$, $\gamma A4_{EC1-6}$, $\gamma A9_{EC1-5}$, $\gamma A9_{EC1-6}$, $\gamma B6_{EC1-3}$, $\gamma B6_{EC1-6}$, $\gamma C3_{EC3-6}$, $\gamma C3_{EC1-6}$). Proteins were diluted to an absorbance at 10 mm and 280 nm of 0.65, 0.43, and 0.23 in channels A, B, and C, respectively. The dilution buffer was used as blank. All samples were run in duplicate at four speeds, the lowest speed was held for 20 hr then four scans with 1 hr interval, the subsequent three speeds were each held for 10 hr followed by four scans with 1 hr interval. The speeds were 9000, 11000, 13000, and 15000 rpm (all EC1–6, EC2–6, and EC1–5 constructs) or 11000, 14000, 17000, and 20000 rpm (all EC1–3, EC1–4, and EC3–6 constructs). Measurements were taken at 25°C, and detection was by UV at 280 nm. Solvent density and protein v-bar at both temperatures were determined using the program SednTerp (Alliance Protein Laboratories, Corte Cancion, Thousand Oaks, CA). For calculation of dimeric $K_d$ and apparent molecular weight, all useful data were used in a global fit, using the program HeteroAnalysis, obtained from University of Connecticut. (www.biotech.uconn.edu/auf). Calculation of the tetramer $K_d$s was done with the program Sedphat (http://www.analyticalultracentrifugation.com/sedphat/index.htm).

## Crystallization and X-ray data collection

Protein crystals were grown using the vapor diffusion method. Crystallization conditions were as follows, with cryo-protectants used given in parentheses: 8% (w/v) PEG8000, 16% ethylene glycol, 20% Morpheus Amino Acids (Molecular Dimensions), 0.1 M Morpheus Buffer System 2 (Hepes/MOPS buffer; Molecular Dimensions) pH 7.0 for $\gamma A1_{EC1-4}$; 11% isopropanol, 50 mM sodium chloride, 0.1 M Hepes pH 7.5 (30% (v/v) ethylene glycol) for $\gamma A8_{EC1-4}$; 10% (w/v) PEG4000, 20% (v/v) glycerol, 30 mM magnesium chloride, 30 mM calcium chloride, 0.1 M Morpheus Buffer System 1 (Mes/Imidazole buffer; Molecular Dimensions) pH 6.5 for $\gamma A9_{EC1-5}$; 8.3% (w/v) PEG8000, 16.7% (v/v) ethylene glycol, 30 mM magnesium chloride, 30 mM calcium chloride, 0.1 M Morpheus Buffer System 2 (Hepes/MOPS buffer; Molecular Dimensions) pH 7.5 for $\gamma B2_{EC1-5}$; 0.1 M Tris-Cl pH 8.5, 0.2 M trimethylamine N-oxide, 3% dextran sulfate sodium salt 5000, 17% (w/v) PEG2000MME (20% (v/v) glycerol) for $\gamma B7_{EC1-4}$ crystal form 1; 0.1 M Tris-Cl pH 8.5, 0.2 trimethylamine N-oxide, 5% (v/v) Jeffamine M-600 pH 7.0, 17% (w/v) PEG2000MME (20% (v/v) PEG400) for $\gamma B7_{EC1-4}$ crystal form 2; 0.1 M ammonium sulfate, 9% (w/v) PEG20000, 18% PEG550MME, 0.1 M Morpheus Buffer System 3 (Tris/Bicine; Molecular Dimensions) pH 8.5 for $\gamma A4_{EC3-6}$; 11.5% (w/v) PEG8000, 23% (v/v) ethylene glycol, 30 mM magnesium chloride, 30 mM calcium chloride, 0.1 M Morpheus Buffer System 1 (Mes/Imidazole buffer; Molecular Dimensions) pH 6.5 for $\gamma B2_{EC3-6}$. X-ray diffraction data was collected at 100K from single crystals at Northeastern Collaborative Access Team (NE-CAT) beamlines 24ID-C and 24ID-E at the Advanced Photon Source, Argonne National Laboratory. All datasets were indexed using XDS (*Kabsch, 2010*) and initially scaled using AIMLESS (*Evans, 2006*; *Evans and Murshudov, 2013*), except the $\gamma A8_{EC1-4}$ data which was indexed with iMOSFLM (*Battye et al., 2011*) and scaled using SCALA (*Evans, 2006*).

### Diffraction anisotropy and pseudosymmetry

The $\gamma A9_{EC1-5}$, $\gamma B7_{EC1-4}$ crystal form 1, and $\gamma A4_{EC3-6}$ diffraction data all showed strong diffraction anisotropy, with much weaker diffraction along a*or b* or both (*Figure 1—figure supplement 2* and *Figure 5—figure supplement 1*). These data were therefore truncated using ellipsoidal limits with using a 3.0 F/sigma cut-off along each of the three principle crystal axes as implemented in the UCLA Diffraction Anisotropy Server (*Strong et al., 2006*). However we did not use the server's default scaling procedure to remove anisotropy from the data in the final rounds of refinement.

Instead an overall anisotropic B-factor was applied to the model by Phenix (*Adams et al., 2010*), as is standard, during refinement to account for the data anisotropy.

The $\gamma B2_{EC3-6}$ diffraction data showed translational pseudosymmetry with a large Patterson peak (60.9% height relative to the origin) at 0.000, 0.000, 0.323. This likely affected the intensity statistics and it is possible this also led to the higher R-values obtained in refinement: Final $R_{work}/R_{free}$ (24.78/27.78%) were higher than is common for a 2.3 Å dataset despite the apparent high quality of the electron density map.

## Crystal structure phasing and refinement

All structures were solved by molecular replacement using Phaser (*McCoy et al., 2007*): $\gamma A1_{EC1-4}$ was solved using the $\gamma A1_{EC1-3}$ structure (PDB: 4ZI9) as a search model; $\gamma A8_{EC1-4}$ was solved using $\gamma A8_{EC1-3}$ (PDB: 4ZPS); $\gamma A9_{EC1-5}$ was solved using EC2–3 of $\gamma A8_{EC1-4}$; $\gamma B7_{EC1-4}$ was solved using ensembles of individual Pcdh EC domains from multiple isoform structures; $\gamma B2_{EC3-6}$ was solved using EC3–5 from the $\alpha 7_{EC1-5}$ structure (PDB: 5DZV); $\gamma A4_{EC3-6}$ was solved using EC3–4 from $\gamma A8_{EC1-4}$, EC5 from $\gamma A9_{EC1-5}$ and EC6 from $\gamma B2_{EC3-6}$; and $\gamma B2_{EC1-5}$ was solved using EC3–5 from $\gamma B2_{EC3-6}$.

Iterative model building using Coot (*Emsley et al., 2010*) and maximum-likelihood refinement using Phenix (*Adams et al., 2010*) was conducted yielding the final refined structures whose statistics are reported in *Figure 1—source data 1* and *Figure 5—source data 1*.

The electron density maps obtained were generally of reasonable quality, however the $\gamma B7_{EC1-4}$ crystal form 2 map had poor density for the bottom half of EC4 in chain B and the neighboring top half of EC1 in chain A. Side chains were not observed in the map for many of the residues in these regions and were therefore not built. The density for EC4 in chain A and EC1 in chain B, including the interfacial regions was much better. The $\gamma A9_{EC1-5}$ map showed poor electron density for EC1, and the $\gamma A4_{EC3-6}$ map showed poor density for EC3. In addition the $\gamma A1_{EC1-4}$, $\gamma A8_{EC1-4}$, and $\gamma B7_{EC1-4}$ crystal form 1 structures were all very low resolution, at 4.2 Å, 3.6 Å, 4.5/4.5/3.6 Å respectively, and therefore many of the side chain positions/rotamers were not clearly defined in the electron density map. We therefore limited our analysis of the interfacial regions of these molecules to looking at which residues were in close proximity rather than the precise atomic arrangements.

## Structure analysis

UCSF Chimera (*Pettersen et al., 2004*) was used to generate unmodeled side chains using the Dunbrack rotamer library (*Dunbrack, 2002*) prior to buried surface area (BSA) calculations. BSAs are given as the change in accessible surface area over both protomers and were calculated using 'Protein interfaces, surfaces and assemblies' service (PISA) at the European Bioinformatics Institute (http://www.ebi.ac.uk/pdbe/prot_int/pistart.html; *Krissinel and Henrick, 2007*). Interdomain angles were calculated using UCSF Chimera. Root mean square deviations over aligned C$\alpha$ atoms between structures were calculated using Pymol (Schrödinger, LLC). Crystal structure figures were made using Pymol.

## Generation of Pcdh isoform sequence conservation logos

Orthologs of the mouse $\gamma$A- and $\gamma$B-Pcdh isoforms were collected from an annotation pipeline link at the NCBI database (*Wheeler et al., 2008*). Blast (*Altschul et al., 1997*) was used to filter out any candidate orthologs with significant similarity to more than one mouse Pcdh isoform. The species for which we identified orthologs of the mouse $\gamma$A- and $\gamma$B-Pcdh isoforms are listed in *Figure 3—source data 1* and *Figure 4—source data 1*. Multiple sequence alignments were generated using Clustal Omega (*Sievers et al., 2011*) and sequence logos were generated using WebLogo3 (*Crooks et al., 2004*).

## Cell aggregation assay to test *trans* binding mutants

A pMax expression construct encoding full-length Pcdh$\gamma$A8 with a C-terminal mCherry-tag was used as described in *Thu et al. (2014)*. Mutants were generated using the Quikchange method (Stratagene). Cell aggregation assays were performed two times as previously described in *Thu et al. (2014)* using K562 cells obtained from ATCC (human leukemia cell line, ATCC CCL243, RRID:CVCL_0004). The cells were mycoplasma free and cell line identity was not verified following purchase.

Briefly, the Pcdh expression constructs were transfected into K562 cells by electroporation using an Amaxa 4D-Nucleofactor (Lonza). After 24 hr, the transfected cells were mixed by shaking for one to three hours. The cells were then imaged with an Olympus fluorescent microscope to determine whether or not they had aggregated.

### Co-transfection assays testing cell surface delivery of α-Pcdhs by other Pcdhs and mutants

Co-transfection assays were performed twice, as previously described in *Thu et al. (2014)* and in a similar manner to the cell aggregation assays described above. C-terminal mCherry-tagged constructs of full length Pcdhα4 or Pcdhα7 were co-transfected with C-terminal mCherry-tagged constructs of various △EC1 Pcdhs and Pcdh mutants into K562 cells by electroporation as described above. Transfected cells were mixed by shaking for 1–3 hr and then imaged to see whether they had aggregated, as described above. Each construct was also transfected into K562 cells alone to confirm that both the △EC1 Pcdhs and the α-Pcdhs could not mediate cell aggregation when expressed alone, as previously observed (*Thu et al., 2014*).

### Accession numbers

Atomic coordinates and structure factors are deposited in the protein data bank with accession codes PDB: 5SZL, 5SZM, 5SZN, 5SZO, 5SZP, 5SZQ, 5SZR, and 5T9T.

## Acknowledgements

We thank Surajit Banerjee, Igor Kourinov, David Neau, and Frank V. Murphy for help with synchrotron data collection at the APS NE-CAT 24-ID-C/E beamlines, supported by NIH P41 GM103403. We thank Seetharaman Jayaraman for help with preliminary data collection for $γB2_{EC3-6}$ at the SSRL BL14.1 beamline. The computing in this project was supported by two National Institutes of Health instrumentation grants (S10OD012351 and S10OD021764) received by the Department of Systems Biology at Columbia University.

## Additional information

### Funding

| Funder | Grant reference number | Author |
|---|---|---|
| Howard Hughes Medical Institute | | Fabiana Bahna<br>Göran Ahlsén<br>Barry Honig |
| National Institutes of Health | R01GM107571 | Tom Maniatis<br>Lawrence Shapiro |
| National Science Foundation | MCB-1412472 | Barry Honig |
| National Institutes of Health | R01GM062270 | Lawrence Shapiro |

The funders had no role in study design, data collection and interpretation, or the decision to submit the work for publication.

### Author contributions

KMG, RR, Conception and design, Acquisition of data, Analysis and interpretation of data, Drafting or revising the article; CAT, Conception and design, Acquisition of data, Analysis and interpretation of data; SM, FB, GA, CR, Acquisition of data, Analysis and interpretation of data; TM, BH, LS, Conception and design, Analysis and interpretation of data, Drafting or revising the article

### Author ORCIDs

Barry Honig, http://orcid.org/0000-0002-2480-6696
Lawrence Shapiro, http://orcid.org/0000-0001-9943-8819

# Additional files

## Major datasets

The following datasets were generated:

| Author(s) | Year | Dataset title | Dataset URL | Database, license, and accessibility information |
|---|---|---|---|---|
| Goodman KM, Bahna F, Mannepalli S, Honig B, Shapiro L | 2016 | Protocadherin gamma A1 extracellular cadherin domains 1-4 | http://www.rcsb.org/pdb/explore/explore.do?structureId=5SZL | Publicly available at the RCSB Protein Data Bank (accession no. 5SZL) |
| Goodman KM, Mannepalli S, Bahna F, Honig B, Shapiro L | 2016 | Protocadherin gamma A8 extracellular cadherin domains 1-4 | http://www.rcsb.org/pdb/explore/explore.do?structureId=5SZM | Publicly available at the RCSB Protein Data Bank (accession no. 5SZM) |
| Goodman KM, Mannepalli S, Bahna F, Honig B, Shapiro L | 2016 | Protocadherin gamma A9 extracellular cadherin domains 1-5 | http://www.rcsb.org/pdb/explore/explore.do?structureId=5SZN | Publicly available at the RCSB Protein Data Bank (accession no. 5SZN) |
| Goodman KM, Mannepalli S, Bahna F, Honig B, Shapiro L | 2016 | Protocadherin Gamma B7 extracellular cadherin domains 1-4 P41212 crystal form | http://www.rcsb.org/pdb/explore/explore.do?structureId=5SZO | Publicly available at the RCSB Protein Data Bank (accession no. 5SZO) |
| Goodman KM, Mannepalli S, Bahna F, Honig B, Shapiro L | 2016 | Protocadherin Gamma B7 extracellular cadherin domains 1-4 P21 crystal form | http://www.rcsb.org/pdb/explore/explore.do?structureId=5SZP | Publicly available at the RCSB Protein Data Bank (accession no. 5SZP) |
| Goodman KM, Mannepalli S, Bahna F, Honig B, Shapiro L | 2016 | Protocadherin gamma A4 extracellular cadherin domains 3-6 | http://www.rcsb.org/pdb/search/structid-Search.do?structureId=5SZQ | Publicly available at the RCSB Protein Data Bank (accession no. 5SZQ) |
| Goodman KM, Mannepalli S, Bahna F, Honig B, Shapiro L | 2016 | Protocadherin Gamma B2 extracellular cadherin domains 3-6 | http://www.rcsb.org/pdb/explore/explore.do?structureId=5SZR | Publicly available at the RCSB Protein Data Bank (accession no. 5SZR) |
| Goodman KM, Mannepalli S, Bahna F, Honig B, Shapiro L | 2016 | Protocadherin Gamma B2 extracellular cadherin domains 1-5 | http://www.rcsb.org/pdb/explore/explore.do?structureId=5T9T | Publicly available at the RCSB Protein Data Bank (accession no. 5T9T) |

The following previously published datasets were used:

| Author(s) | Year | Dataset title | Dataset URL | Database, license, and accessibility information |
|---|---|---|---|---|
| Goodman KM, Bahna F, Mannepalli S, Honig B, Shapiro L | 2016 | Protocadherin alpha 4 extracellular cadherin domains 1-4 | http://www.rcsb.org/pdb/explore/explore.do?structureId=5DZW | Publicly available at the RCSB Protein Data Bank (accession no. 5DZW). |
| Goodman KM, Bahna F, Mannepalli S, Honig B, Shapiro L | 2016 | Protocadherin beta 8 extracellular cadherin domains 1-4 | http://www.rcsb.org/pdb/explore/explore.do?structureId=5DZY | Publicly available at the RCSB Protein Data Bank (accession no. 5DZY). |
| Goodman KM, Mannepalli S, Bahna F, Honig B, Shapiro L | 2016 | Protocadherin beta 6 extracellular cadherin domains 1-4 | http://www.rcsb.org/pdb/explore/explore.do?structureId=5DZX | Publicly available at the RCSB Protein Data Bank (accession no. 5DZX). |
| Goodman KM, Bahna F, Honig B, Shapiro L | 2016 | Protocadherin alpha 7 extracellular cadherin domains 1-5 | http://www.rcsb.org/pdb/explore/explore.do?structureId=5DZV | Publicly available at the RCSB Protein Data Bank (accession no. 5DZV). |

| Nicoludis JM, Lau S-Y, Scharfe CPI, Marks DS, Weihofen WA, Gaudet R | 2015 | Structure of mouse clustered PcdhgA1 EC1-3 | http://www.rcsb.org/pdb/explore/explore.do?structureId=4Zl9 | Publicly available at the RCSB Protein Data Bank (accession no. 4Zl9). |
|---|---|---|---|---|
| Nicoludis JM, Vogt BE, Gaudet R | 2016 | Structure of human clustered protocadherin gamma B3 EC1-4 | http://www.rcsb.org/pdb/explore/explore.do?structureId=5K8R | Publicly available at the RCSB Protein Data Bank (accession no. 5K8R). |

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
