## [Decision Letter]

Congratulations, we are pleased to inform you that your article, "Functional implications of γ-Protocadherin structural diversity", is, in principle, acceptable for publication in *eLife*.

Before publication, please address the Minor Comments raised by reviewers 1 and 2, and also provide as a supplementary figure the analytical ultracentrifugation data showing the wt and mutant V562D in Figure 6.

*Reviewer #1 (General assessment and major comments (Required)):*

The authors report extensive structural and biochemical data on protocadherin trans (i.e. cell-cell) dimerization specificity critical for specification of neuronal contacts. These dimers form by antiparallel association of the first four EC domains, such that EC1 and EC2 of one molecule interact with EC4 and 3, respectively, of the partner. With this paper there are now representatives from each of the clustered protocadherin subfamilies, enabling a thorough bioinformatic analysis of specificity. They show that the gamma A and B isoforms are distinct subfamilies. In the last part of the paper, they use structural data and mutagenesis to define the cis-dimer interface mediated by EC6, and suggest possible roles of cis-dimers giving rise to higher order assemblies that might contribute to combinatorial specificity amongst the protocadherins. Overall this work represents a significant advance in understanding protocadherin function.

Recently Gaudet and colleagues (Nicoludis et al.) published short *eLife* paper with a single structure of a gammaB3 EC1-4 fragment and used this, along with published examples from other family members, to conclude that specificity is encoded solely in the EC2:3 interface, whereas the EC1:4 provides needed affinity but not specificity. This was based in part on the claim that a general, non-specific hydrophobic interface is present in all isoforms. The present paper gives compelling arguments that there is indeed specificity in this interface, including that they had previously published domain swap experiments demonstrating specificity in the EC1:4 interface. The authors note the presence of a salt bridge in this interface that cannot form in mismatched pair. Could it be that the salt bridge is energetically inconsequential to dimer formation, and the family-wise specificity comes from the lack of electrostatic repulsion?

More broadly, the fact that the EC1:4 interface is so variable and splayed in a couple of structures suggests that it is quite plastic. There is not an obvious correlation with crystallization conditions, although it is interesting that the crystals with isopropanol show this separation (as well as one copy in the gammaA1 structure, which is at the lowest pH of those that show any dimerization). It would be helpful to offer some speculation as to the origin of these differences, and also to clarify if the alternative 1-4 interaction in the gammaA8 lattice is distinct. Do the authors think that the EC1:4 interface is a transient interaction rather than stable?

*Reviewer #1 (Minor Comments):*

1) In the last paragraph of the subsection “Arrangement of γB-Pcdh trans dimers”: it is not clear until later in the paper that the gammaB3 structure is an outlier; for example, the gammaA's show similar rmsds within that family. The authors should clarify the basis of this statement here or at least point the reader to the analysis later in the paper.

2) Given the severe anisotropy that is documented, it would be useful to put the refined anisotropic B tensor values into crystallographic tables.

*Reviewer #2 (General assessment and major comments (Required)):*

The article presents several crystal structures of gamma-protocadherins and supports them with AUC data and cell aggregation assays. The manuscript is focused on identifying structural determinants of binding specificity among protocadherin isoforms for both trans and cis interactions. Beyond that, the manuscript does not claim to provide much further insight into self-avoidance mechanisms in mammalian neurons, although the cis interaction data may help build useful future models of Pcdh adhesion complexes formed between neuronal processes.

The crystal structures presented appear to have been solved properly, given repeatedly low resolution, anisotropic diffraction and one case of pseudo-translational symmetry. Their ugly *Rmerge* statistics can be easily explained by the issues they report with their datasets, and the final models of the structures have decent geometry. I am not concerned about the crystallography. For the cell aggregation results, the representative images look very clear and convincing, although by its nature the method is only semi-quantitative (and they did not quantitate it).

The authors have come up with a few interesting conclusions. Their first claim is that EC1 to 4 (not only EC2 and 3) are involved in determining trans binding specificity. They also argue that the previous γ-B3 structure (Nicoludis, 2016) is non-native (or partly artefactual). I found both of these to be convincing given their data. For the pairs/groups of residues they focus as specificity determinants (such as R340-E41-K338), the structure based data is very suggestive, yet all such claims would have benefited from some experimental data (such as some sort of swap mutagenesis). Such risky experiments need not be performed for a revision. Similar claims on structural bases for binding preferences within classes, or homophilic vs. heterophilic are also strongly suggestive, although always hard to prove.

The last part of the article on cis binding is more novel, although lack of a dimeric crystal structure for EC5-6 is unfortunate. The test used to assess cis binding is also quite indirect: the idea is, cis dimerization causes trafficking to the cell surface, which then leads to trans interactions, and hence cell aggregation. While I am convinced, I believe a more direct assay would put to rest most concerns readers have – clearly not easy to do, and not necessary for a possible revision. One specific and mild concern I had here is the folding of some of the mutants tested for identifying the cis interface, which never made it to the cell surface (a common outcome for misfolded proteins), and they couldn't express two of these three mutants for AUC experiments either (another sign of misfolding). In other words, lack of surface trafficking might represent not lack of cis binding but also misfolding of the mutants.

Overall, while I like the structures and the analysis, I am split on whether *eLife* is a good venue for publishing these results. The trans binding results are not as novel, and lack of a cis complex structure decreases my enthusiasm for the manuscript. Yet, the structures are useful and a significant forward step in resolving Pcdh complex specificity and topology issues.

*Reviewer #2 (Minor Comments):*

In Discussion, subsection “Implications for neuronal recognition”, clarify unlike "previously suggested", "the repertoire composition of cis-dimers is predicted to be non-uniform" is one of manuscript's new findings. Otherwise, that paragraph is confusing.

*Reviewer #2 (Additional data files and statistical comments):*

No concerns.

*Reviewer #3 (General assessment and major comments (Required)):*

This manuscript contains a very extensive analysis of the structural nature of cell recognition by clustered protocadherins of the nervous system. It concerns a very important problem because this recognition mechanism plays a major role in determining how neurons interact physically with each other. Although this is not the first contribution of this type to this field, it provides a lot of valuable detailed information that will help us understand the overall mechanisms. The authors conclusions about the location of specificity determinants differs somewhat from those of a recent study by Nicoludis et al., providing a compelling and alternative view.

The core of the study relies on comparisons between a very large number of crystallographic structure determinations of various members of these protocadherin families. The authors confirm trans interactions that have been reported in previous studies, but the wealth of structural information in this study provides insights into the detailed nature of interaction specificity. Some surprising minimal trans interactions were also observed, e.g. with γA8EC1-4, but the authors provide mutagenesis analysis along with cell aggregation activity to support the structure. A major hypothesis based on the authors' previous biochemical analyses and physiological functions described in the field, that these protocadherins form cis parallel hetero-dimers via C-terminal EC domains, failed to be observed in the crystal structures. Nonetheless, the authors analyzed putative cis interaction regions via mutagenesis and cell biological assays, involving co-transport of heterologous partners to the cell surface, to provide insights into the likely cis dimer interfaces.

Overall, this extensive analysis of multiple protocadherins using a variety of molecular, biochemical and cellular approaches, provides a wealth of information that contributes significantly to the field.

*Reviewer #3 (Minor Comments):*

None.

---

## [Author Response]

*Reviewer #1 (General assessment and major comments (Required)):*

*[…] Recently Gaudet and colleagues (Nicoludis et al.) published short eLife paper with a single structure of a gammaB3 EC1-4 fragment and used this, along with published examples from other family members, to conclude that specificity is encoded solely in the EC2:3 interface, whereas the EC1:4 provides needed affinity but not specificity. This was based in part on the claim that a general, non-specific hydrophobic interface is present in all isoforms. The present paper gives compelling arguments that there is indeed specificity in this interface, including that they had previously published domain swap experiments demonstrating specificity in the EC1:4 interface. The authors note the presence of a salt bridge in this interface that cannot form in mismatched pair. Could it be that the salt bridge is energetically inconsequential to dimer formation, and the family-wise specificity comes from the lack of electrostatic repulsion?*

*More broadly, the fact that the EC1:4 interface is so variable and splayed in a couple of structures suggests that it is quite plastic. There is not an obvious correlation with crystallization conditions, although it is interesting that the crystals with isopropanol show this separation (as well as one copy in the gammaA1 structure, which is at the lowest pH of those that show any dimerization). It would be helpful to offer some speculation as to the origin of these differences, and also to clarify if the alternative 1-4 interaction in the gammaA8 lattice is distinct. Do the authors think that the EC1:4 interface is a transient interaction rather than stable?*

We thank the reviewer for his/her in-depth assessment of our work. With respect to the differences among EC1:EC4 interactions in γA-Pcdhs, we have not yet been able to assign the determinants responsible for the apparent plasticity of the EC1:EC4 interactions in γA-Pcdhs. However, the fact that we observe alternate arrangements only for γA-Pcdhs suggests that this is related to a property of the γA subfamily. We have now made it clear that, despite the alternate arrangements observed crystallographically, AUC experiments with γA1 and γA4 show that the presence of all four domains is required for binding (Table 1). While there is clear malleability of the EC1:EC4 interaction in the γA subfamily, we have no evidence that this interaction is transient.

We have also now clarified that the alternative EC1:EC4 interaction observed in the γA8 crystal lattice is indeed a distinct interaction from the EC1:EC4 interaction that occurs in the fully engaged EC1–4 *trans* dimer structures such γA1_EC1–4_ chains A&B. Since we have no evidence to suggest γA8_EC1–4_ makes additional interactions beyond the *trans* dimer interaction, we believe this alternative EC1:EC4 interaction is most likely a non-biological crystal lattice contact.

*Reviewer #1 (Minor Comments):*

*1) In the last paragraph of the subsection “Arrangement of γB-Pcdh trans dimers”: it is not clear until later in the paper that the gammaB3 structure is an outlier; for example, the gammaA's show similar rmsds within that family. The authors should clarify the basis of this statement here or at least point the reader to the analysis later in the paper.*

We understand the confusing aspect of this comparison. The γA subfamily, which is highly structurally plastic, shows high RMSDs among structures within the family. However, all other Pcdh subfamilies – which did not display such plasticity – have low RMSDs in comparisons between family members. This is part of the reason why RMSD identifies γB3 as an outlier within the γB subfamily. Importantly, it is the combination of this observation, along with the apparent disruption of the EC2:EC3 interface, which identifies γB3 as an outlying structure. We have now clarified that the discussion in this paragraph is limited to the γB subfamily structures.

*2) Given the severe anisotropy that is documented, it would be useful to put the refined anisotropic B tensor values into crystallographic tables.*

We thank the reviewer, and have now included anisotropic B tensor values in the crystallographic tables.

*Reviewer #2 (General assessment and major comments (Required)):*

*[…] Overall, while I like the structures and the analysis, I am split on whether eLife is a good venue for publishing these results. The trans binding results are not as novel, and lack of a cis complex structure decreases my enthusiasm for the manuscript. Yet, the structures are useful and a significant forward step in resolving Pcdh complex specificity and topology issues.*

We thank the reviewer for his/her thorough evaluation of our paper. With respect to the folding of mutants tested for identifying the *cis* interface, we chose residues that were surface exposed in the EC6-domain-containing structures reported here. While we understand the reviewers’ concern, we think it is unlikely that mis-folding is responsible for the lack of surface expression for these two mutants.

*Reviewer #2 (Minor Comments):*

*In Discussion, subsection “Implications for neuronal recognition”, clarify unlike "previously suggested", "the repertoire composition of cis-dimers is predicted to be non-uniform" is one of manuscript's new findings. Otherwise, that paragraph is confusing.*

We thank the reviewer. We agree that this language was confusing, and have now clarified it in the revised text.

*Reviewer #3 (General assessment and major comments (Required)):*

*[…] Overall, this extensive analysis of multiple protocadherins using a variety of molecular, biochemical and cellular approaches, provides a wealth of information that contributes significantly to the field.*

We thank the reviewer for their positive assessment of our paper.